# Symmetric Pruning in Quantum Neural Networks

**Xinbiao Wang**[1,2]**, Junyu Liu**[3,4,5,6]**, Tongliang Liu**[7]**, Yong Luo**[1,8,9]**, Yuxuan Du**[2,†]**, Dacheng Tao**[2]

[1]Institute of Artificial Intelligence, School of Computer Science, Wuhan University, China    [2]JD Explore Academy
[3]Pritzker School of Molecular Engineering, The University of Chicago    [4]Chicago Quantum Exchange
[5]Kadanoff Center for Theoretical Physics    [6]qBraid Co.    [7]Sydney AI Centre, The University of Sydney,
[8]National Engineering Research Center for Multimedia Software, School of Computer Science, Institute of Artificial
 Intelligence and Hubei Key Laboratory of Multimedia and Network Communication Engineering, Wuhan University,
China    [9]Hubei Luojia Laboratory, Wuhan, China    †: corresponding author

## Abstract

Many fundamental properties of a quantum system are captured by its Hamiltonian and ground state. Despite the significance, ground states preparation (GSP) is classically intractable for most large-scale Hamiltonians. Quantum neural networks (QNNs), which exert the power of modern quantum machines, have emerged as a leading protocol to conquer this issue. As such, the performance enhancement of QNNs becomes the core in GSP. Empirical evidence showed that QNNs with handcraft symmetric ansätze generally experience better trainability than those with asymmetric ansätze, while theoretical explanations remain vague. To fill this knowledge gap, here we propose the effective quantum neural tangent kernel (EQNTK) and connect this concept with over-parameterization theory to quantify the convergence of QNNs towards the global optima. We uncover that the advance of symmetric ansätze attributes to their large EQNTK value with low effective dimension, which requests few parameters and quantum circuit depth to reach the over-parameterization regime permitting a benign loss landscape and fast convergence. Guided by EQNTK, we further devise a symmetric pruning (SP) scheme to automatically tailor a symmetric ansatz from an over-parameterized and asymmetric one to greatly improve the performance of QNNs when the explicit symmetry information of Hamiltonian is unavailable. Extensive numerical simulations are conducted to validate the analytical results of EQNTK and the effectiveness of SP.

## 1 Introduction

The law of quantum mechanics advocates that any quantum system can be described by a Hamiltonian, and many important physical properties are reflected by its ground state. For this reason, the ground state preparation (GSP) of Hamiltonians is the key to understanding and fabricating novel quantum matters. Due to the intrinsic hardness of GSP (Poulin & Wocjan, 2009; Carleo et al., 2019), the required computational resources of classical methods are unaffordable when the size of Hamiltonian becomes large. Quantum computers, whose operations can harness the strength of quantum mechanics, promise to tackle this problem with potential computational merits. In the noisy intermediate-scale quantum (NISQ) era (Preskill, 2018), quantum neural networks (QNNs) (Farhi & Neven, 2018; Cong et al., 2019; Cerezo et al., 2021a) are leading candidates toward this goal. The building blocks of QNNs, analogous to deep neural networks, consist of variational ansätze (also called parameterized quantum circuits) and classical optimizers. In order to enhance the power of QNNs in GSP, great efforts have been made to design advanced ansätze with varied circuit structures (Peruzzo et al., 2014; Wecker et al., 2015; Kandala et al., 2017).

Despite the achievements aforementioned, recent progress has shown that QNNs may suffer from severe trainability issues when the circuit depth of ansätze is either shallow or deep.

Namely, for the deep ansätze, the magnitude of the gradients exponentially decays with the increased system size (McClean et al., 2018; Cerezo et al., 2021b). This phenomenon, dubbed the barren plateau, hints at the difficulty of optimizing deep QNNs, where an exponential runtime is necessitated for convergence. The wisdom to alleviate barren plateaus is exploiting shallow ansätze to accomplish learning tasks (Grant et al., 2019; Skolik et al., 2021; Zhang et al., 2020; Pesah et al., 2021), while the price to pay is incurring another serious trainability issue—convergence (Boyd & Vandenberghe, 2004; Du et al., 2021). The trainable parameters may get stuck into sub-optimal local minima or saddle points with high probability because of the unfavorable loss landscape (Anschuetz, 2021; Anschuetz & Kiani, 2022). Orthogonal to these negative results, several studies pointed out that when the depth of ansätze becomes overwhelmingly deep and surpasses a critical point, the over-parameterized QNNs embrace a benign landscape and permit fast convergence towards good local minima (Kiani et al., 2020; Wiersema et al., 2020; Larocca et al., 2021b). Nevertheless, the criteria to reach such a critical point is stringent, i.e., the number of parameterized gates or the circuit depth scales exponentially with the problem size, which hurdles the application of over-parameterized QNNs in practice.

Empirical evidence sheds new light on exploiting over-parameterized QNNs to tackle GSP. QNNs with symmetric ansätze only demand a polynomial number of trainable parameters and the circuit depth with the problem size to reach the over-parameterized region and achieve a fast convergence rate (Herasymenko & O'Brien, 2021; Gard et al., 2020; Zheng et al., 2021; 2022; Shaydulin & Wild, 2021; Mernyei et al., 2022; Marvian, 2022; Meyer et al., 2022; Larocca et al., 2022; Sauvage et al., 2022). A common feature of these symmetric ansätze is capitalizing on the symmetric properties underlying the problem Hamiltonian to shrink the solution space and facilitate seeking near-optimal solutions. Unfortunately, current symmetric ansätze are inapplicable to a broad class of Hamiltonians whose symmetry is implicit, since their constructions rely on the explicit information for the symmetry of Hamiltonians. Besides, it is unknown whether the symmetry contributes to lowering the critical point to reach the over-parameterization regime.

Here we fill the above knowledge gap from both theoretical and practical aspects. Concretely, we develop a novel notion—effective quantum neural tangent kernel (EQNTK) to capture the training dynamic of various ansätze via their **effective dimension**. In doing so, we expose that compared with the asymmetric

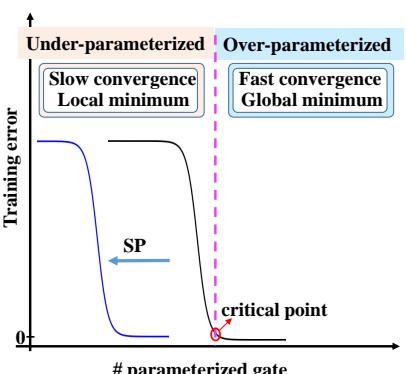

Figure 1: **The critical point of the over-parameterized regime.** When the number of parameters is beyond the critical point (the red circle), the training error exponentially converges to a nearly global minimum. Symmetric ansätze (the blue curve) require few parameters to reach the critical point over the asymmetric ansätze.

ansätze, the symmetric ansätze possess dramatically *lower effective dimensions* and the required number of parameters and circuit depth to reach the over-parameterization may polynomially scale with the problem size (see Fig. 1 for an intuition). By leveraging EQNTK, we next prove that when the condition of over-parameterization is satisfied, the trainable parameters of QNNs with symmetric ansätze can **exponentially converge** to the global optima with the increased iterations. Taken together, our analysis recognizes that over-parameterized QNNs with symmetric ansätze is a possible solution toward large-scale GSP tasks. Envisioned by EQNTK and pruning techniques in deep neural networks (Han et al., 2015; Blalock et al., 2020; Frankle et al., 2020; Wang et al., 2022), we further devise a **symmetric pruning scheme (SP)** to *automatically tailor a symmetric ansatz from an over-parameterized and asymmetric one* with the enhanced trainability and applicability. Conceptually, SP continuously eliminates the redundant quantum gates from the given asymmetric ansatz and correlates parameters to assign different types of symmetries on the slimmed ansatz. In this way, SP generates a modest-depth symmetric ansatz with a fast convergence guarantee and thus improves the hardware efficiency. Extensive simulations on many-body physics and combinatorial problems validate the theory of EQNTK and the

efficacy of SP. These results deepen our understating about how to merge symmetry with over-parameterization theory and indicate the signification of designing symmetric ansätze.

**Contributions.** We summarize our main contributions as follows:

1. We propose the notion of EQNTK to quantify the training dynamics of QNNs with symmetric ansätze, which reconciles **the QNTK theory with the symmetry of the problem Hamiltonian** (see Sec. 2.2). As shown in Fig. 1, since the training dynamic between symmetric and asymmetric ansätze is evidently disparate, our results provide a deep understanding towards QNNs with symmetric ansätze, especially for unraveling how the structure information effects the convergence rate.

2. Our key **technical contribution** is achieving a tighter convergence bound of QNNs with various symmetric ansätze (see Theorem 2 and Lemma 1). Particularly, our bound yields $\gamma = O(\text{poly}(LK, d_{\text{eff}}^{-1}))$, where $LK$ is the number of parameters and $d_{\text{eff}}$ is the effective dimension. The comparison with prior results is summarized in Table 1. Our results not only greatly reduce the threshold in reaching over-parameterization but promise an improved convergence rate. These two conclusions are indispensable in applying over-parameterized QNNs to solve practical problems.

|   | Larocca et al. (2021b) | Anschuetz (2021) | You et al. (2022) | Liu et al. (2022b) | **Our results** |
|---|---|---|---|---|---|
| C | $\Omega(\text{poly}(d_{\text{eff}}))$ | $\Omega(\exp(n))$ | $\mathcal{O}(\text{poly}(d_{\text{eff}}))$ | $\mathcal{O}(\exp(n))$ | $\mathcal{O}(\text{poly}(d_{\text{eff}}))$ |
| T | ✗ | ✗ | $\mathcal{O}(\log(d_{\text{eff}})\log(\frac{1}{\epsilon}))$ | $\mathcal{O}(\frac{4^n \log(\frac{1}{\varepsilon})}{LK})$ | $\mathcal{O}(\frac{d_{\text{eff}}^2 \log(\frac{1}{\varepsilon})}{LK})$ |

Table 1: **A comparison of the convergence rate for over-parameterized QNNs.** The label 'C' and 'T' refers to the critical point and $\epsilon$-convergence rate respectively. The label "✗" denotes that the paper did not study certain regimes. Note that the achieved results in Ref. (Larocca et al., 2021b) do not exhibit how the problem Hamiltonian effects $d_{\text{eff}}$.

3. Our last contribution is devising SP, an automatic scheme to identify the implicit symmetries of the problem Hamiltonian and utilize them to design a symmetric ansatz (see Section 3). An attractive feature of SP is its flexibility, where any heuristic that has the ability to capture certain symmetries of the problem Hamiltonian can be seamlessly embedded into SP to further boost its performance.

## 2 Effective QNTK allows an improved convergence of QNNs

Here we establish foundations about why symmetric ansätze have the ability to enhance the trainability of QNNs in ground state preparation (GSP) tasks. To do so, we propose a novel concept—effective quantum neural tangent kernel (EQNTK), to reconcile the QNTK theory with the symmetry of the problem Hamiltonian. Attributed to EQNTK, we uncover that the advance of symmetric ansätze originates from their ability to dramatically decrease the over-parameterization threshold. For elucidating, we first interpret the necessary backgrounds in Sec. 2.1 and then present our main theoretical results in Sec. 2.2.

### 2.1 Problem setup

**Ground state preparation.** Given an $n$-qubit Hamiltonian $H \in \mathbb{C}^{2^n \times 2^n}$, GSP aims to find the eigenvector $|\psi^*\rangle \in \mathbb{C}^{2^n}$ (i.e., the ground state) of $H$ corresponding to its minimum eigenvalue. For any $n$-qubit state $|\psi\rangle$, the variational principle ensures $\langle\psi^*| H |\psi^*\rangle \leq \langle\psi| H |\psi\rangle$ and the equality is satisfied iff $|\psi\rangle = |\psi^*\rangle$. Since the dimension of $|\psi^*\rangle$ scales exponentially with $n$, GSP is classically intractable for a large $n$.

**Quantum neural networks.** A QNN can be described by a triplet $(|\psi_0\rangle, U(\boldsymbol{\theta}), H)$. When it is applied to solve GSP, an ansatz $U(\boldsymbol{\theta})$ (i.e., a parameterized unitary) prepares a variational state $|\psi(\boldsymbol{\theta})\rangle = U(\boldsymbol{\theta}) |\psi_0\rangle$ with a fixed input state $|\psi_0\rangle$. The parameters $\boldsymbol{\theta}$ are optimized by minimizing the loss function

$$\mathcal{L}(\boldsymbol{\theta}) = \frac{1}{2} \left( \langle\psi_0| U(\boldsymbol{\theta})^\dagger H U(\boldsymbol{\theta}) |\psi_0\rangle - E_0 \right)^2, \tag{1}$$

where $E_0 = \langle \psi^* | H | \psi^* \rangle$ refers to the ground state energy of $H$. The optimization follows an iterative manner, i.e., the classical optimizer continuously leverages the output of the quantum circuits to update $\boldsymbol{\theta}$ and the update rule is $\boldsymbol{\theta}^{(t+1)} = \boldsymbol{\theta}^{(t)} - \eta \partial \mathcal{L}(\boldsymbol{\theta}^{(t)})/\partial \boldsymbol{\theta}$, where $\eta$ refers to the learning rate. See Appendix A for details.

**Remark.** We adopt $E_0$ to facilitate the convergence analysis and our results cover general loss functions where $E_0$ is replaced by $C \in \mathbb{R}$ with $C \leq E_0$. See Appendix B for details.

**Constructions of (symmetric) ansätze**. The power of QNNs depends on the employed ansatz $U(\boldsymbol{\theta})$. A general form of $U(\boldsymbol{\theta})$ covering many typical ansätze such as Hamiltonian variational ansatz (HVA) and hardware efficient ansatz (HEA) (Bharti et al., 2022; Qian et al., 2021) yields

$$U(\boldsymbol{\theta}) = \prod_{\ell=1}^{L} U_\ell(\boldsymbol{\theta}_\ell), \quad U_\ell(\boldsymbol{\theta}_\ell) = \prod_{k=1}^{K} e^{-iG_k \boldsymbol{\theta}_{\ell k}}, \tag{2}$$

where $L$ refers to the layer number, $\boldsymbol{\theta} = (\boldsymbol{\theta}_1, \cdots, \boldsymbol{\theta}_L) \in \Theta \subseteq \mathbb{R}^{LK}$ is trainable parameters living in the parameter space $\Theta$, $\boldsymbol{\theta}_\ell = (\boldsymbol{\theta}_{\ell 1}, \cdots, \boldsymbol{\theta}_{\ell K})$ is trainable parameters at the $\ell$-th layer, and $\mathcal{A} = \{G_1, \cdots, G_K\}$ is a set of Hermitian traceless operators called an *ansatz design*. Given $\Theta$ and $\mathcal{A}$, a set of ansätze forms a subgroup of $SU(2^n)$ with $\mathcal{U}_\mathcal{A} = \cup_{L=0}^{\infty}\{U(\boldsymbol{\theta}) : \boldsymbol{\theta} \in \Theta\}$. The difference of ansätze originates from the varied $\Theta$ and $\mathcal{A}$. Given a Hermitian matrix $\Sigma$, the ansatz $U(\boldsymbol{\theta})$ is said to be symmetric with respect to $\Sigma$ if each element in $\mathcal{U}_\mathcal{A}$ is commutable with $\Sigma$. Mathematically, denote $\Sigma = \sum_{j=1}^{p} \sum_{k=1}^{s_j} \lambda_j \boldsymbol{v}_{jk}$ where $\lambda_j$ is the eigenvalue with $\lambda_i \neq \lambda_j$ for $i \neq j$, $\boldsymbol{v}_{jk}$ is the corresponding eigenvector, and $\sum_{j=1}^{p} s_j = 2^n$. The explicit form of $\Sigma$ leads to a direct sum decomposition $\mathcal{H} = \oplus_{j=1}^{p} V_j$ of the quantum state space, where $V_j$ is the invariant subspace spanned by the eigenvectors $\{\boldsymbol{v}_{j1}, \cdots, \boldsymbol{v}_{js_j}\}$.

**Convergence of QNNs**. A crucial metric to assess the performance of different QNNs is the $\epsilon$-convergence rate towards the global minimum $\mathcal{L}(\boldsymbol{\theta}^*)$ with $\boldsymbol{\theta}^* = \min_{\boldsymbol{\theta} \in \Theta} \mathcal{L}(\boldsymbol{\theta})$.

**Definition 1** ($\epsilon$-convergence). *A QNN instance $(|\psi_0\rangle, U(\boldsymbol{\theta}), H)$ achieves an $\epsilon$-convergence if the trained parameters after $T$ iterations $\boldsymbol{\theta}^{(T)}$ satisfy $\mathcal{L}(\boldsymbol{\theta}^{(T)}) \leq \epsilon$ with $\epsilon \in \mathbb{R}$.*

This quantity measures the distance between the estimated and the optimal loss values, which can be derived via the quantum neural tangent kernel (QNTK)

$$Q^{(t)} = \nabla \varepsilon_t^\top \nabla \varepsilon_t, \tag{3}$$

where $\varepsilon_t = \langle \psi_0 | U(\boldsymbol{\theta}^{(t)})^\dagger H U(\boldsymbol{\theta}^{(t)}) | \psi_0 \rangle - E_0$ denotes the residual training error and $\nabla \varepsilon_t$ is the gradients of $\varepsilon_t$ with respect to $\boldsymbol{\theta}$. The following theorem describes the $\epsilon$-convergence of the over-parameterized QNN with an arbitrary ansatz.

**Theorem 1** (Liu et al. (2022b)). *Following notations in Eqns. (1)-(3), when $U(\boldsymbol{\theta})$ matches the Haar distribution up to the fourth moment, the number of parameters satisfies $LK \gg 1$, and the learning rate $\eta \ll 1$, the training dynamics of a QNN instance $(|\psi_0\rangle, U(\boldsymbol{\theta}), H)$ yields*

$$\varepsilon_t \approx (1 - \eta \bar{Q})^t \varepsilon_0 \approx e^{-\gamma t} \varepsilon_0. \tag{4}$$

*where $\gamma = \eta \bar{Q}$ is the indicator of the decay rate and $\bar{Q} = \mathcal{O}(LK \operatorname{Tr}(H^2)/4^n)$ refers to the expectation of $Q$ on Haar average.*

It indicates that the critical point to reach the over-parameterization region is $|\boldsymbol{\theta}| \sim \mathcal{O}(4^n/(\eta \operatorname{Tr}(H^2)))$. In this setting, the exponent in Eqn. (4) meets $\gamma \sim \mathcal{O}(1)$ and promises an exponential convergence. Besides, Eqn. (4) hints that the convergence rate of QNNs is continuously enhanced by increasing the value of QNTK, which can be achieved by growing the number of parameters or decreasing the system size.

## 2.2 Effective Quantum Neural Tangent Kernel

The exponential scaling behavior with the number of qubits $n$ in Theorem 1 causes the realization of the over-parameterized QNNs to be impractical for large systems. Moreover, the corresponding convergence rate is **independent of the refined structure information** of ansätze. This contradicts with the empirical evidence such that symmetric ansätze outperform asymmetric ansätze with fast convergence in training QNNs. It thus highly demands carrying out new theories to stress these issues.

Here we propose a novel concept—effective QNTK (EQNTK) to resolve the above dilemma and exhibit how symmetry improves the trainability of QNNs. As shown in Fig. 2, given a QNN $(|\psi_0\rangle, U(\boldsymbol{\theta}), H)$ whose input state $|\psi_0\rangle$ and the ground state $|\psi^*\rangle$ live in the same subspace $V^* \subset \mathbb{C}^{2^n}$ and $V^*$ is invariant with the employed symmetric ansatz $U(\boldsymbol{\theta})$, the **training dynamics** of $|\psi(\boldsymbol{\theta})\rangle = U(\boldsymbol{\theta})|\psi_0\rangle$ can be exactly captured by $V^*$, a much smaller space than the whole state space. Suppose that the state space $\mathcal{H}$ under the symmetric ansatz design $\mathcal{A}$ can be decomposed into $\mathcal{H} = \oplus_{j=1}^p V_j$ and there exists $j^* \in [p]$ such that $V_{j^*} = V^*$ includes the input state $|\psi_0\rangle$, the ground state $|\psi^*\rangle$, and all possible variational states $\{|\psi(\boldsymbol{\theta})\rangle | \boldsymbol{\theta} \in \Theta\}$. Then the dynamics of $|\psi(\boldsymbol{\theta})\rangle$ can be derived by the dimension of this subspace $d_{\text{eff}} = |V^*|$, dubbed the *effective dimension* of the QNN.

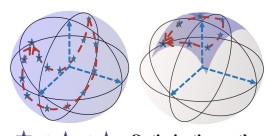

★–▸★–▸★  **Optimization path**

Figure 2: **Training dynamics of QNNs with symmetric ansätze.** The left and right panels illustrate the dynamic of variational states corresponding to the asymmetric and symmetric ansätze, respectively. The shadow region means the solution space, which is the whole Hilbert space for asymmetric ansatz, and the restricted invariant subspace for symmetric ansatz.

**Definition 2** (Effective dimension). *Consider a QNN instance $(|\psi_0\rangle, U(\boldsymbol{\theta}), H)$ with symmetric ansatz design $\mathcal{A}$. Suppose $V^*$ is the invariant subspace covering $|\psi_0\rangle$, $\{|\psi(\boldsymbol{\theta})\rangle | \boldsymbol{\theta} \in \Theta\}$, and $|\psi^*\rangle$. Then the effective dimension of this QNN is $d_{\text{eff}} = |V^*|$. The projection on this subspace is defined as $\Pi = PP^\dagger$, where $P \in \mathbb{C}^{d \times d_{\text{eff}}}$ is an arbitrary set of orthonormal basis.*

As a result, for symmetric ansätze, $Q^{(t)}$ and $\bar{Q}$ in Theorem 1 should be controlled by $d_{\text{eff}}$ instead of $2^n$. This integration of the effective dimension transforms QNTK in Eqn. (3) to EQNTK, which reduces the threshold to reach over-parameterization and accelerates the convergence (see Fig. 1). The following theorem establishes the convergence theory of QNNs with symmetric ansätze under EQNTK, whose proof is deferred to Appendix C.

**Theorem 2.** *Consider the QNN instance $(|\psi_0\rangle, U(\boldsymbol{\theta}), H)$ with the effective dimension $d_{\text{eff}}$. Following notations in Eqns. (1)-(3), when the distribution of $U(\boldsymbol{\theta})$ constrained to the invariant subspace with projection $\Pi = PP^\dagger$ matches the Haar distribution up to the fourth moment, the number of parameters satisfies $LK \gg 1$, the learning rate $\eta \ll 1$, and denoting $H^* = PHP^\dagger$, the training dynamics of a QNN instance $(|\psi_0\rangle, U(\boldsymbol{\theta}), H)$ yields*

$$\varepsilon_t \approx (1 - \eta\bar{Q}_S)^t \varepsilon_0 \approx e^{-\gamma t}\varepsilon_0. \tag{5}$$

*where $\bar{Q}_S = \mathcal{O}(LK \operatorname{Tr}((H^*)^2)/d_{\text{eff}}^2)$ refers to the expectation of EQNTK $Q_S$ on Haar average.*

The above results indicate that when the number of trainable parameters scales with $LK \sim \mathcal{O}(d_{\text{eff}}^2/(\eta \operatorname{Tr}((H^*)^2)))$, the adopted symmetric ansatz reaches the over-parameterization regime. Compared with QNTK, the reduction of parameters in the order of $(2^n/d_{\text{eff}})^2$ not only ensures the practical utility of over-parameterized QNNs, but also explains the empirical observations that symmetric ansätze require fewer parameters to reach the critical point than that of the asymmetric ansätze. More importantly, unlike prior results arguing that the trainability can always be improved by over-parameterization, our bound suggests that involving more parameters beyond the critical point may degrade the convergence since the underlying symmetry may be broken and the effective dimension $d_{\text{eff}}$ could be large.

**Remark.** (i) The derived EQNTK can be used to diagnose the barren plateaus of QNNs. Particularly, the quantity $Q_S/(LK)$ amounts to the variance of the gradient whose average is zero under the 4-design assumption. In other words, when the number of parameters $LK$ is fixed, a large EQNTK value is preferred to avoid barren plateaus. (ii) The effective dimension can be quantified by other metrics beyond the dimension of the invariant subspace. An alternative is the dynamical Lie algebra (DLA) (Larocca et al., 2021b), which measures the controllability of the quantum system. The following lemma shows the convergence of QNNs with symmetric ansätze under this measure, whose proof is given in Appendix D.

**Proposition 1** (Informal). *Following notations in Theorem 2, denote the dynamical Lie algebra of the QNN instance $(|\psi_0\rangle, U(\boldsymbol{\theta}), H)$ as $\mathfrak{g}$, and assume that the number of parameters $LK \gg 1$ and the learning rate $\eta \ll 1$. If there exists an invariant subspace $V_{\mathfrak{g}}$ with dimension*

$d_{\mathfrak{g}}$ *under the DLA $\mathfrak{g}$ including the input state $|\psi_0\rangle$ and the ground state $|\psi^*\rangle$, the DLA-based EQNTK $Q_D$ corresponding to the ansatz $U(\boldsymbol{\theta})$ leads to the training dynamics*

$$\varepsilon_t \approx (1 - \eta \bar{Q}_D)^t \varepsilon_0 \approx e^{-\gamma t} \varepsilon_0. \qquad (6)$$

*where $\bar{Q}_D = \mathcal{O}(LK \operatorname{Tr}((H^*)^2)/d_{\mathfrak{g}}^2)$ refers to the expectation of $Q_D$ on Haar average.*

## 3 SYMMETRICAL PRUNING WITH EQNTK

Beyond analyzing the convergence rate, another ad-hoc topic in GSP is designing advanced ansätze to improve the trainability of QNNs. Although over-parameterization and contemporary symmetric ansätze partially address this problem, both of them have evident caveats. The former may request exponential parameters to satisfy the condition of over-parameterization, while the latter requires explicit information for the symmetries of the problem Hamiltonian. To compensate for these deficiencies, here we devise **symmetrical pruning (SP)**, an automatic scheme to design symmetric ansätze with the enhanced trainability of QNNs. Conceptually, SP distills a symmetric over-parameterized ansatz from an asymmetric over-parameterized ansatz. Supported by the EQNTK theory, the extracted ansatz is resource-friendly in implementation since it holds a small effective dimension and only needs a few trainable parameters to compass the over-parameterization.

---

**Algorithm 1:** Symmetric pruning (SP)

**Input** : Problem Hamiltonian
$\widetilde{H} = (\sum_{j=1}^{q} \alpha_j H_j) \otimes \mathbb{I}^{\otimes m}$, the
ansatz design $\mathcal{A}$ and the
parameter space $\Theta$ in Eqn. (2).
Step 1. Initialize an over-parameterized
and asymmetric ansatz via $\mathcal{A}$ and $\Theta$;
Step 2. Symmetry identification:
2-1. Remove the gates on wires
corresponding to the redundant part of $\widetilde{H}$
in $\mathcal{A}$, i.e., $\mathbb{I}^{\otimes m}$.
2-2. Remove the gates such that the
pruned ansatz design $\mathcal{A}_{\mathrm{pr}} = \{H_1, \cdots, H_q\}$.
2-3. Assign the spatial symmetry of $\mathcal{A}_{\mathrm{pr}}$
by correlating some parameters and
obtain $\Theta_{\mathrm{pr}} \subseteq \Theta$.
**Output:** Pruned ansatz design $\mathcal{A}_{\mathrm{pr}}$ and
parameter space $\Theta_{\mathrm{pr}}$.

---

The Pseudo code of SP is summarized in Alg. 1 and its schematic illustration is shown in Fig. 3. Suppose the problem Hamiltonian is $\widetilde{H} = H \otimes \mathbb{I}^{\otimes m}$, where $H = \sum_{j=1}^{q} \alpha_j H_j$, $\alpha_j$ is the real coefficient and $H_j$ is the tensor product of Pauli matrices on $n$ qubits, SP builds the symmetric ansatz of $\widetilde{H}$ with two primary steps, i.e., initialization and symmetry identification. The initialization step is choosing an initial over-parameterized QNN by setting down the ansatz design $\mathcal{A}$ and the parameter space $\Theta$. Note that $\mathcal{A}$ should contain all Pauli terms in $H$ and $\Theta$ should ensure an $\epsilon$-convergence of QNNs, e.g., a possible choice is adopting a sufficient deep hardware efficient ansatz. Next, the symmetry identification step iteratively discovers the system symmetry, structure symmetry, and spatial symmetry, which is completed by three sub-steps. Step 2-1 symmetrically prunes the qubit wires. That is, all qubit gates interact with the redundant part of $\widetilde{H}$, i.e., the identity term $\mathbb{I}^{\otimes m}$, are removed. Step 2-2 symmetrically prunes the structure. This step drops the parameterized single-qubit gates and the two-qubit gates so that the pruned ansatz design $\mathcal{A}_{\mathrm{pr}}$ can be block diagonalized under the projection on the eigenspace of $H = \sum_{j=1}^{q} \alpha_j H_j$. A possible solution is setting $\mathcal{A}_{\mathrm{pr}} = \{H_1, \cdots, H_q\}$ and the pruned ansatz $U_{\mathrm{pr}}(\boldsymbol{\theta})$ takes the form of Eqn. (2) with $U_\ell(\boldsymbol{\theta}_\ell) = \Pi_{k=1}^{q} e^{-iH_k \boldsymbol{\theta}_{\ell k}}$. Step 2-3 correlates symmetric parameters to demystify the spatial symmetry of $H$, which is accomplished by a heuristic related to identifying the graph automorphism group (Stoichev, 2019). See Appendix E for more details.

**Remark.** (i) We emphasize that although both SP and the pruning techniques used in deep neural networks orient to remove redundant parameters and (quantum) neurons, they are fundamentally different. This is because classical pruning methods generally leverage the magnitude of weights or the gradient information to recognize such redundancy, which is impermissible in QNNs (refer to Appendix F for elaborations). (ii) SP is a flexible framework. Besides three symmetric properties in Alg. 1, SP can effectively integrate with other symmetry identification methods in the second step.

**: Gates to be removed**     **: Correlated parameters**

Figure 3: **Schematic of symmetric pruning.** The proposed symmetric pruning (SP) distills the symmetric ansatz from a given asymmetric ansatz, completed by removing the redundant gates (highlighted by the red dashed boxes) and correlating the parameters in the gate respecting the spatial symmetry (highlighted by the solid boxes with the same color).

## 4 EXPERIMENTS

We carry out numerical simulations to explore the theoretical properties of EQNTK and validate the effectiveness of the SP scheme in GSP. Two typical problem Hamiltonians in many-body physics and combinatorial optimization are considered. The omitted details are postponed to Appendix G.

**Problem Hamiltonian.** Let us first recap the two problem Hamiltonians.

**1) Transverse-field Ising model.** Transverse-field Ising model (TFIM) has been employed to explore many interesting quantum systems. An $n$-qubit Hamiltonian of 1D TFIM with an open boundary condition is defined as $H_{\text{TFIM}} = -\sum_{j=1}^{n-1} \sigma_j^z \sigma_{j+1}^z - h \sum_{j=1}^{n} \sigma_j^x$, where $\sigma_j^\mu$ denotes the $\mu$-Pauli matrix (with $\mu = x, z$) acting on the $j$-th qubit, and $h$ is the strength of the transverse field. For simplicity, we set $h = 1$ in the following simulations.

2) **Maximum cut.** Maximum cut (MaxCut) problem aims to partition the set of nodes $V$ in a graph $G = (V, E)$ into two parts such that the number of edges spanning multiple parts is maximized. The MaxCut problem can be recast to GSP. Namely, the objective of an $n$-node graph is encoded by an $n$-qubit Hamiltonian $H_{\text{MC}} = \frac{1}{2} \sum_{(u,v) \in E} (I - \sigma_u^z \sigma_v^z)$ and the optimal solution corresponds to the ground state of $H_{\text{MC}}$ as formulated in Eqn. (1). Here we focus on the Erdos-Renyi graphs, which are generated by randomly connecting any pair nodes among $n$ nodes with probability $p = 0.6$.

To verify the effectiveness of SP, the above problem Hamiltonians are modified as an $(n+m)$-qubit Hamiltonian $H = H_M \otimes \mathbb{I}^m (H_M = H_{\text{TFIM}}, H_{\text{MC}})$ with $n = 6$ and $m = 2$.

**Initialization of QNNs.** The hardware efficient ansatz (HEA) with the form of Eqn. (2) is used as the initial ansatz, which is over-parameterized and asymmetric. The layer number is set as $L \in \{4, 6, 8 \cdots, 28\}$ and $L \in \{4, 6, 10, \cdots, 36\}$ for TFIM and MaxCut, respectively.

For each problem Hamiltonian, the input state is set as $|\psi_0\rangle = |\mathbf{0}\rangle$. The parameters $\boldsymbol{\theta}$ are uniformly sampled from the uniform distribution $[-\pi, \pi]$. The variational ansatz is trained by the Adam optimizer where the learning rate is 0.001 and the rest hyper-parameters follow the default settings. The training of QNNs stops when the loss value is less than $10^{-8}$ or when the change in the loss function is less than $10^{-8}$ three times in a row. The maximum number of iterations is set as $T = 10000$. The $\epsilon$ value in Definition 1 is set as $10^{-5}$ for both TFIM and MaxCut. Each setting is repeated with 5 times to collect the statistical results.

**Evaluation metrics.** We utilize three metrics to assess the convergence rate of QNNs, i.e., (1) the loss value $\mathcal{L}(\boldsymbol{\theta}^{(T)})$ at the convergence stage; (2) the number of iteration steps $T(\epsilon) \leq T$ required to achieve the $\epsilon$-convergence; (3) the minimum number of parameterized gates required to achieve $\epsilon$-convergence, which can also be interpreted as the threshold to achieve the over-parameterization regime. Additionally, we record the norm of the gradient at the initialization to verify the effectiveness of EQNTK in predicting the convergence.

### 4.1 SIMULATION RESULTS

**EQNTK at initialization.** Fig. 4(b) and Fig. 5(b) show that each symmetric pruning step in Alg. 1 (i.e., the sub-steps 2-1, 2-2, and 2-3 refer to 'SP1', 'SP2', 'SP3', respectively) constantly improves the squared norm of gradients (i.e., the EQNTK value $Q_S$) for both

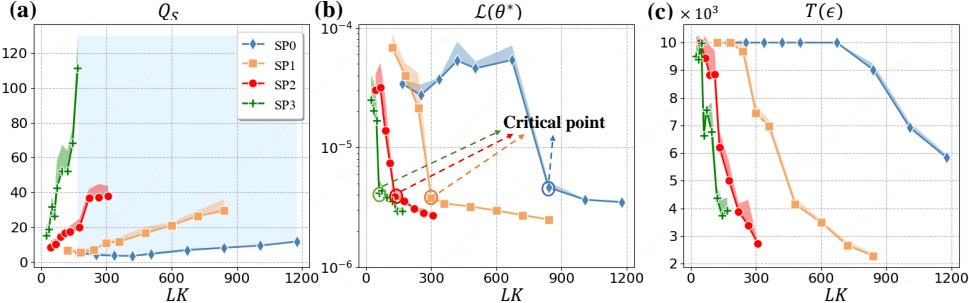

Figure 4: **Results for TFIM model under symmetric pruning.** The panels (a)-(c) plot the EQNTK value $Q_S$ at initialization, the loss value after convergence $\mathcal{L}(\boldsymbol{\theta}^{(T)})$, and the iteration number to achieve the $\epsilon$-convergence $T(\varepsilon)$ versus the number of parameters $LK$, respectively. The labels 'SP0'-'SP3' refer to the initial ansatz, the pruned ansatz after system symmetric pruning, structure symmetric pruning, and spatial symmetric pruning, respectively.

TFIM and MaxCut. This implicates that SP is capable of accelerating convergence of QNNs and alleviating barren plateaus based on Theorem 2. Notably, the EQNTK $Q_S$ of the pruned ansatz (labeled by 'SP3') is improved by 20 (or 14) times compared to the initial over-parameterized ansatz (labeled by 'SP0') in the problem of TFIM (or MaxCut).

**Critical point of QNNs.** Fig. 4(c) and Fig. 5(c) illustrate that when the number of parameters surpasses a threshold, QNNs experience a computational phase transition where the loss value $\mathcal{L}(\boldsymbol{\theta}^{(T)})$ at the convergence stage sharply drops by an order of the magnitude. Moreover, the minimum number of parameters required to reach the over-parameterization regime, highlighted by the 'critical point', is dramatically reduced by SP. Specifically, the number of parameters of naive QNNs at the critical point in TFIM (or MaxCut), i.e., labeled by 'SP0', scales exponentially with the system size. By contrast, it is gradually reduced from 800 (or 1000) to 300 (or 300) after SP1, then to 120 (or 150) after SP2, and finally to 50 (or 100) after SP3, which scales polynomially with the system size and is resource-friendly for modern quantum devices (see Appendix G for hardware efficiency analysis).

**Convergence of QNNs.** Fig. 4(d) and Fig. 5(d) reflect that SP dramatically improves the convergence of QNNs. In the common over-parameterization regime of 'SP1'–'SP3' (i.e., $LK \geq 300$), the total iterations required to achieve $\epsilon$-convergence can be reduced by up to 6000 steps for TFIM and 5000 steps for MaxCut. For the same ansatz design, increasing the number of parameters linearly improves the convergence, which echoes with Theorem 2.

**EQNTK and trainability of QNNs.** Fig. 4 and Fig. 5 indicate the relation between the EQNTK value and the trainability of QNNs. That is, the convergence rate and the number of parameters around the critical point decrease with the increased EQNTK value. In MaxCut, when the number of parameters $LK \approx 450$ reaches the over-parameterization regime in the cases of 'SP1'–'SP3', the corresponding EQNTK value yields $Q_1 : Q_2 : Q_3 \approx 1 : 2 : 4$ and the iteration steps $T(\epsilon)$ follows $T_1 : T_2 : T_3 \approx 4 : 2 : 1$ ($T_j$ and $Q_j$ refer to $T(\epsilon)$ and $Q_S$ in 'SP$j$' with $j \in [3]$). A similar phenomenon is observed in the task of TFIM. These results accord with Theorem 2 such that EQNTK can guide the trainability of QNNs.

## 5    RELATED WORK

Prior literature related to our work can be cast into two categories: the trainability theories of QNNs and the design of symmetric ansätze.

**Trainability of QNNs.** McClean et al. (2018) first discovered the barren plateau of QNNs. Since then, a line of research is uncovering intrinsic reasons leading to this phenomenon. Current progress has revealed that these reasons include high entanglement of QNNs (Marrero et al., 2021), the used global measurements (Cerezo et al., 2021b), and the presence of noise (Wang et al., 2021). To mitigate barren plateaus, two popular ways are adopting

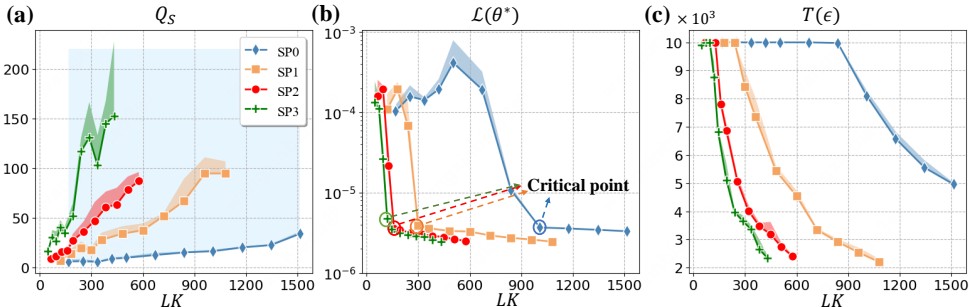

Figure 5: **Results for MaxCut under SP.** The notations are identical to those in Fig. 4.

shallow QNNs with local measurements (Uvarov & Biamonte, 2021; Pesah et al., 2021; Du et al., 2022a; Zhang et al., 2020) and correlating parameters (Volkoff & Coles, 2021).

Another crucial line of research is investigating the convergence rate of QNNs. Several empirical studies have observed that over-parameterized QNNs promise faster convergence, and the trained parameters are near-optimal (Kiani et al., 2020; Wiersema et al., 2020; Zhang & Cui, 2020). Afterward, initial attempts have been made to theoretically explain the superiority of over-parameterized QNNs. Specifically, Larocca et al. (2021b) and Anschuetz (2021) separately leveraged the tools of dynamical Lie algebra and random matrix theory to quantify the critical point of over-parameterized QNNs; You et al. (2022) extended the result of Xu et al. (2018) to the quantum regime and proved the exponential convergence rate of over-parameterized QNNs; Liu et al. (2022c;b;d;a) proposed the quantum neural tangent kernel to exhibit an exponential convergence rate of over-parameterized QNNs. A common caveat of prior literature is that their convergence analysis either requires an exponential circuit depth for reaching over-parameterization or omits the factor of the circuit depth. By contrast, EQNTK greatly reconciles the harsh requirement to reach over-parameterization and allows a tighter convergence rate for the symmetric ansatz.

**Ansätze with symmetric properties.** Previous studies focus on unearthing inherent symmetry behind the problem Hamiltonian to design problem-specific ansätze. The mainstream approaches contain arranging the layout of ansätze (Liu et al., 2019; Seki et al., 2020; Gard et al., 2020; Zheng et al., 2021; 2022), correlating trainable parameters (Shaydulin et al., 2021; Shaydulin & Wild, 2021; Sauvage et al., 2022), and utilizing results from the geometric deep learning (Shaydulin et al., 2021; Shaydulin & Wild, 2021; Sauvage et al., 2022; Meyer et al., 2022), where the symmetry comes from the training data. Compared with asymmetric ansätze, these ansätze enable better trainability in GSP. However, none of the previous proposals can identify the implicit symmetry of the problem Hamiltonian. Moreover, although there is numerical evidence that symmetric ansätze can accelerate convergence, theoretical analysis is still rare. EQNTK readily compensates these issues, which provides an efficient measure to compare the trainability of various ansätze and allows an automatic method (symmetric pruning) to design symmetrical ansatz with fast convergence.

## 6   CONCLUSIONS

In this study, we investigate the training performance of QNNs for the GSP problem by developing a novel tool—EQTNK, which is capable of capturing the training dynamics of various ansätze via their effective dimension. We prove that a symmetric ansatz design with a small effective dimension enables an improved trainability of QNNs, including alleviating the barren plateaus and reducing the number of parameters and the circuit depth required to reach the over-parameterization regime. Besides, we propose a novel symmetric pruning algorithm to automatically extract the symmetric ansatz from an over-parameterized and asymmetric ansatz. Empirical results confirm the effectiveness of SP. A future research direction is extending the results of EQNTK from GSP to the regime of machine learning and exploring whether over-parameterized QNNs can simultaneously attain good trainability and generalization (Abbas et al., 2021; Du et al., 2022b; Caro et al., 2022).

## 7 ACKNOWLEDGEMENTS

Yong Luo was supported by the Special Fund of Hubei Luojia Laboratory under Grant 220100014 and the National Natural Science Foundation of China under Grant 62276195. Tongliang Liu was partially supported by Australian Research Council Projects IC-190100031, LP-220100527, DP-220102121, and FT-220100318.

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

## A    Optimization of QNNs in GSP

In this section, we separately elaborate the elementary notions in quantum computing, the preliminary of Hamiltonian and the ground state preparation (GSP), and the optimization strategy of QNNs in the task of GSP.

**Basics of quantum computation.** The elementary unit of quantum computation is qubit (or quantum bit), which is the quantum mechanical analogue of a classical bit. A qubit is a two-level quantum-mechanical system described by a unit vector in the Hilbert space $\mathbb{C}^2$. In Dirac notation, a qubit state is defined as $|\phi\rangle = c_0 |0\rangle + c_1 |1\rangle \in \mathbb{C}^2$ where $|0\rangle = [1,0]^\top$ and $|1\rangle = [0,1]^T$ specify two unit bases and the coefficients $c_0, c_1 \in \mathbb{C}$ yield $|c_0|^2 + |c_1|^2 = 1$. Similarly, the *quantum state* of $n$ qubits is defined as a unit vector in $\mathbb{C}^{2^n}$, i.e., $|\psi\rangle = \sum_{j=1}^{2^n} c_j |e_j\rangle$, where $|e_j\rangle \in \mathbb{R}^{2^n}$ is the computational basis whose $j$-th entry is 1 and other entries are 0, and $\sum_{j=1}^{2^n} |c_j|^2 = 1$ with $c_j \in \mathbb{C}$. Besides Dirac notation, the density matrix can be used to describe more general qubit states. For example, the density matrix of the state $|\psi\rangle$ is $\rho = |\psi\rangle \langle\psi| \in \mathbb{C}^{2^n \times 2^n}$, where $\langle\psi| = |\psi\rangle^\dagger$ refers to the complex conjugate transpose of $|\psi\rangle$. For a set of qubit states $\{p_j, |\psi_j\rangle\}_{j=1}^m$ with $p_j > 0$, $\sum_{j=1}^m p_j = 1$, and $|\psi_j\rangle \in \mathbb{C}^{2^n}$ for $j \in [m]$, its density matrix is $\rho = \sum_{j=1}^m p_j \rho_j$ with $\rho_j = |\psi_j\rangle \langle\psi_j|$ and $\text{Tr}(\rho) = 1$.

A *quantum gate* is an unitary operator which can evolve a quantum state $\rho$ to another quantum state $\rho'$. Namely, an $n$-qubit gate $U \in \mathcal{U}(2^n)$ obeys $UU^\dagger = U^\dagger U = I_{2^n}$, where $\mathcal{U}(2^n)$ refers to the unitary group in dimension $2^n$. Typical single-qubit quantum gates include the Pauli gates, which can be written as Pauli matrices:

$$X = \begin{bmatrix} 0 & 1 \\ 1 & 0 \end{bmatrix}, \quad Y = \begin{bmatrix} 0 & -i \\ i & 0 \end{bmatrix}, \quad Z = \begin{bmatrix} 1 & 0 \\ 0 & -1 \end{bmatrix}. \tag{7}$$

The more general quantum gates are their corresponding rotation gates $R_X(\theta) = e^{-i\frac{\theta}{2}X}, R_Y(\theta) = e^{-i\frac{\theta}{2}Y}$, and $R_Z(\theta) = e^{-i\frac{\theta}{2}Z}$ with a tunable parameter $\theta$, which can be written in the matrix form as

$$R_X(\theta) = \begin{bmatrix} \cos\frac{\theta}{2} & -i\sin\frac{\theta}{2} \\ -i\sin\frac{\theta}{2} & \cos\frac{\theta}{2} \end{bmatrix}, R_Y(\theta) = \begin{bmatrix} \cos\frac{\theta}{2} & -\sin\frac{\theta}{2} \\ \sin\frac{\theta}{2} & \cos\frac{\theta}{2} \end{bmatrix}, R_Z(\theta) = \begin{bmatrix} e^{-i\frac{\theta}{2}} & 0 \\ 0 & e^{i\frac{\theta}{2}} \end{bmatrix}. \tag{8}$$

They are equivalent to rotating a tunable angle $\theta$ around $x$, $y$, and $z$ axes of the Bloch sphere, and recovering the Pauli gates $X$, $Y$, and $Z$ when $\theta = \pi$. Moreover, a multi-qubit gate can be either an individual gate (e.g., CNOT gate) or a tensor product of multiple single-qubit gates.

The *quantum measurement* refers to the procedure of extracting classical information from the quantum state. It is mathematically specified by a Hermitian matrix $H$ called the *observable*. Applying the observable $H$ to the quantum state $|\psi\rangle$ yields a random variable whose expectation value is $\langle\psi| H |\psi\rangle$.

**Hamiltonian and GSP**. In quantum computation, a *Hamiltonian* is a Hermitian matrix that is used to characterize the evolution of a quantum system or as an observable to extract the classical information from the quantum system. Specifically, under the Schrödinger equation, a quantum gate has the mathematical form of $U = e^{-itH}$, where $H$ is a Hermitian matrix, called the Hamiltonian of the quantum system, and $t$ refers to the evolution time of the Hamiltonian. Typical single-qubit Hamiltonians include the Pauli matrices defined in Eqn. (7). As a result, the evolution time $t$ refers to the tunable parameter $\theta$ in Eqn. (8). Any single-qubit Hamiltonian can be decomposed as the linear combination of Pauli matrices, i.e., $H = a_1 I + a_2 X + a_3 Y + a_4 Z$ with $a_j \in \mathbb{C}$. In the same way, a multi-qubit Hamiltonian is denoted by $H = \sum_{j=1}^{4^n} a_j P_j$, where $P_j \in \{I, X, Y, Z\}^{\otimes n}$ is the tensor product of Pauli matrices. In quantum chemistry and quantum many-body physics, the Hermitian matrix describing the quantum system to be solved is denoted as the *problem Hamiltonian*.

When taking the problem Hamiltonian as the observable, the quantum state $|\psi^*\rangle$ is said to be the *ground state* of problem Hamiltonian $H$ if the expectation value $\langle \psi^* | H | \psi^* \rangle$ takes the minimum eigenvalue of $H$, which is called the *ground energy*. GSP refers to preparing the ground state of the problem Hamiltonian. A popular protocol for GSP is to employ a parameterized unitary $U(\boldsymbol{\theta})$ to prepare a variational quantum state $|\psi(\boldsymbol{\theta})\rangle = U(\boldsymbol{\theta}) |\psi_0\rangle$ with a fixed input state $|\psi_0\rangle$ and then optimize the parameters $\boldsymbol{\theta}$ by minimizing a predefined loss function such as the Eqn. (1).

**Optimization of QNNs.** The optimization of the loss function $\mathcal{L}(\boldsymbol{\theta})$ in Eqn. (1) can be completed by gradient-based methods. A plethora of optimizers has been designed to estimate the optimal parameters $\boldsymbol{\theta}^* = \min_{\boldsymbol{\theta}} \mathcal{L}(\boldsymbol{\theta})$. Here we introduce the implementation of the first-order gradient-based optimizer for self-consistency. Refer to Cerezo et al. (2021a) for a comprehensive review.

Based on Eqn. (2), the trainable parameters of QNNs are denoted by $\boldsymbol{\theta} = (\boldsymbol{\theta}_1^\top, \cdots, \boldsymbol{\theta}_L^\top)^\top$ with $\boldsymbol{\theta}_\ell = (\theta_{\ell 1}, \cdots, \theta_{\ell K})^T$, where the subscript '$\ell k$' refers to the $k$-th parameter of the $\ell$-th layer $U_\ell$ for $\forall k \in [K]$ and $\forall \ell \in [L]$. Recall the loss function takes the form

$$\mathcal{L}(\boldsymbol{\theta}) = \frac{1}{2} \left( \langle \psi_0 | U(\boldsymbol{\theta})^\dagger H U(\boldsymbol{\theta}) | \psi_0 \rangle - E_0 \right)^2,$$

and the corresponding update rule at the $t$-th iteration $\forall t \in [T]$ is

$$\boldsymbol{\theta}^{(t+1)}$$
$$= \quad \boldsymbol{\theta}^{(t)} - \eta \frac{\partial \mathcal{L}(\boldsymbol{\theta}^{(t)})}{\partial \boldsymbol{\theta}}$$
$$= \quad \boldsymbol{\theta}^{(t)} - \eta \left( \langle \psi_0 | U(\boldsymbol{\theta}^{(t)})^\dagger H U(\boldsymbol{\theta}^{(t)}) | \psi_0 \rangle - E_0 \right) \frac{\partial \left( \langle \psi_0 | U(\boldsymbol{\theta}^{(t)})^\dagger H U(\boldsymbol{\theta}^{(t)}) | \psi_0 \rangle - E_0 \right)}{\partial \boldsymbol{\theta}},$$

where $\eta$ refers to the learning rate. The derivative in the last equality can be calculated via the parameter shift rule Mitarai et al. (2018). Mathematically, the derivative with respect to the parameter $\theta_{\ell k}$ for $\forall \ell \in [L]$ and $\forall k \in [K]$ is

$$\frac{\partial \left( \langle \psi_0 | U(\boldsymbol{\theta})^\dagger H U(\boldsymbol{\theta}) | \psi_0 \rangle - E_0 \right)}{\partial \theta_{\ell k}}$$
$$= \quad \frac{1}{2 \sin \alpha} \left[ \left( \langle \psi_0 | U(\boldsymbol{\theta}^+)^\dagger H U((\boldsymbol{\theta}^+) | \psi_0 \rangle - E_0 \right) - \left( \langle \psi_0 | U((\boldsymbol{\theta}^-)^\dagger H U((\boldsymbol{\theta}^-) | \psi_0 \rangle - E_0 \right) \right],$$

where $\boldsymbol{\theta}^+ = \boldsymbol{\theta} + \alpha \boldsymbol{e}_{\ell k}$, $\boldsymbol{\theta}^- = \boldsymbol{\theta} - \alpha \boldsymbol{e}_{\ell k}$, $\boldsymbol{e}_{\ell k}$ is the unit vector along the $\theta_{\ell k}$ axis and $\alpha$ can be any real number but the multiple of $\pi$ because of the diverging denominator.

## B EQUIVALENT TRAINING DYNAMICS UNDER THE MEAN SQUARE ERROR LOSS

In the main text, to facilitate the convergence analysis, the loss function $\mathcal{L}(\boldsymbol{\theta})$ in Eqn. (1) adopts the term $E_0$, as the ground state energy of the problem Hamiltonian. Here we elucidate how to extend our results to a more general loss function in which $E_0$ is replaced

by any $C \in \mathbb{R}$ with $C \leq E_0$. More specifically, the general mean square error loss function is defined as

$$\mathcal{L}(\boldsymbol{\theta}, C) = \frac{1}{2} \left( \langle \psi_0 | U(\boldsymbol{\theta})^\dagger H U(\boldsymbol{\theta}) | \psi_0 \rangle - C \right)^2 \equiv \frac{1}{2} \varepsilon(\boldsymbol{\theta}, C)^2, \tag{9}$$

where

$$\varepsilon(\boldsymbol{\theta}, C) = \langle \psi_0 | U(\boldsymbol{\theta})^\dagger H U(\boldsymbol{\theta}) | \psi_0 \rangle - C$$

refers to the training error associated with $C$. For clarity, we denote the training error at the $t$-th iteration as $\varepsilon_t(\boldsymbol{\theta}^{(t)}, C)$. Given two loss functions $\mathcal{L}(\boldsymbol{\theta}, C)$ and $\mathcal{L}(\boldsymbol{\theta}, C')$ with $C, C' \leq E_0$, their convergence behavior or training dynamics is said to be equivalent if the variational quantum state $|\psi(\boldsymbol{\theta})\rangle$ converges to a same quantum state with the same convergence rate. More concretely, for the same initial state $|\psi^{(0)}\rangle$, the evolved state $|\psi^{(t)}\rangle$ at each iteration $t$ for $\forall t \in [T]$ is the same.

The following lemma indicates the equivalent training dynamic of QNNs under the loss functions $\mathcal{L}(\boldsymbol{\theta}, C)$ and $\mathcal{L}(\boldsymbol{\theta}, C')$ with $C, C' \leq E_0$.

**Lemma 1.** *Under the framework of the quantum neural tangent kernel in Eqn. (3), given any loss function $\mathcal{L}(\boldsymbol{\theta}, C)$ in Eqn. (9) with $C \leq E_0$, QNN obeys the same convergence rate with $\mathcal{L}(\boldsymbol{\theta}, E_0)$ and the optimized variational quantum state $|\psi(\boldsymbol{\theta})\rangle$ converges to the ground state of the problem Hamiltonian $H$.*

*Proof of Lemma 1.* In the same manner with Eqn. (3), the QNTK of the loss function in Eqn. (9) can be denoted by

$$Q_C = \nabla \varepsilon(\boldsymbol{\theta}, C)^\top \nabla \varepsilon(\boldsymbol{\theta}, C).$$

According to the explicit form of $\varepsilon(\boldsymbol{\theta}, C)$ in Eqn. (9), the QNTK $Q_C$ and $Q_{C'}$ is the same for any given constant $C$ and $C'$ as $\nabla \varepsilon(\boldsymbol{\theta}, C) = \nabla \varepsilon(\boldsymbol{\theta}, C')$. For this reason, in the following, we use $Q$ to refer the QNTK with any constant $C$.

Recall the results of Theorem 1, i.e., in the case of $C = E_0$, the training error $\varepsilon(\boldsymbol{\theta}, E_0)$ decays as

$$\varepsilon_t(\boldsymbol{\theta}^{(t)}, E_0) \approx e^{-\eta Q t} \varepsilon_0(\boldsymbol{\theta}^{(0)}, E_0).$$

Moreover, due to $\varepsilon_t(\boldsymbol{\theta}^{(t)}, E_0) = \varepsilon_t(\boldsymbol{\theta}^{(t)}, C) + (C - E_0)$ for $\forall t \in [T]$, the training error of QNNs under the loss $\mathcal{L}(\boldsymbol{\theta}, C)$ is

$$\varepsilon_t(\boldsymbol{\theta}^{(t)}, C) + (C - E_0) \approx e^{-\eta Q t} \left( \varepsilon_0(\boldsymbol{\theta}^{(t)}, C) + (C - E_0) \right). \tag{10}$$

Since the right-hand side tends to be zero with a sufficiently large number $t$, this suggests $\varepsilon_t(\boldsymbol{\theta}^{(t)}, C) + (C - E_0) \approx 0$. In other words, $\varepsilon(\boldsymbol{\theta}, C)$ converges to the minimal value of $E_0 - C$ with the decay rate $\eta Q$. Supported by the variational principle, the optimized variational quantum state $U(\boldsymbol{\theta}^{(T)}) | \psi_0 \rangle$ at the converging stage is exactly the ground state in which the corresponding energy estimates $E_0$. □

## C  PROOF OF THEOREM 2

The proof of Theorem 2 employs the following two lemmas whose proofs are given in the subsequent two subsections.

**Lemma 2** (Adapted from You et al. (2022), Lemma D.1). *Following Definition 2, let $\mathcal{U}_A$ be a matrix subgroup of $SU(d)$ where each element in $\mathcal{U}_A$ commutes with a Hermitian matrix $\Sigma$. The corresponding direct decomposition is denoted by $V = \sum_{j=1}^P V_j$ with projection $\Pi_j$. Let $V^*$ be the subspace of interest which includes the input state $|\psi_0\rangle$ and ground state $|\psi^*\rangle$. Denote $\Pi^* = P^\dagger P$ as the projection on $V^*$. Then for any Hermitian $W$ and any unitary matrix $U$ in the group $\mathcal{U}_A$, we have*

$$\Pi^* U W U^\dagger \Pi^* = \Pi^* U \Pi \, \Pi^* W \Pi^* \, \Pi^* U^\dagger \Pi^*. \tag{11}$$

**Lemma 3.** *Following notations in Lemma 2, denote*

$$U_{-,\ell k} \equiv \prod_{\ell'=1}^{\ell-1} U_{\ell'}(\boldsymbol{\theta}_{\ell'}) \prod_{k'=1}^{k-1} e^{-i\boldsymbol{\theta}_{\ell k'} G_{k'}}, \quad U_{+,\ell k} \equiv \prod_{k'=k}^{K} e^{-i\boldsymbol{\theta}_{\ell k'} G_{k'}} \prod_{\ell'=\ell+1}^{L} U_{\ell'}(\boldsymbol{\theta}_{\ell'}), \tag{12}$$

*the EQNTK takes the form*

$$Q_S = -\sum_{\ell=1}^{L}\sum_{k=1}^{K} \left\langle \psi_0^* \Big| (U_{+,\ell k}^*)^\dagger \left[ G_k^*, (U_{-,\ell k}^*)^\dagger H^* U_{-,\ell k}^* \right] U_{+,\ell k}^* \Big| \psi_0^* \right\rangle^2, \tag{13}$$

*where* $|\psi_0^*\rangle = P|\psi^*\rangle$ *and* $A^* = PAP^\dagger$ *with* $A \in \{U_{+,\ell k}, G_k, U_{-,\ell k}, H\}$.

*Proof of Theorem 2.* Following the gradient descent optimizer in Eqn. (1) with the learning rate $\eta \leq 1$, the change of the training error of QNN can be expressed as

$$\text{d}\varepsilon = \sum_{\ell,k} \frac{\partial \varepsilon}{\partial \boldsymbol{\theta}_{\ell k}} \text{d}\boldsymbol{\theta}_{\ell k} = -\eta \sum_{\ell,k} \frac{\partial \varepsilon}{\partial \boldsymbol{\theta}_{\ell k}} \frac{\partial \varepsilon}{\partial \boldsymbol{\theta}_{\ell k}} \varepsilon = -\eta Q_S \varepsilon. \tag{14}$$

where $\text{d}\varepsilon = \varepsilon_{t+1} - \varepsilon_t$ and $\text{d}\boldsymbol{\theta}_{\ell k} = \boldsymbol{\theta}_{\ell k}^{(t+1)} - \boldsymbol{\theta}_{\ell k}^{(t)}$, the second equality comes from the update rule with $\text{d}\boldsymbol{\theta}_{\ell k} = \eta\varepsilon\partial\varepsilon/\partial\boldsymbol{\theta}_{\ell k}$, and the third equality uses the definition of QNTK in Eqn. (3).

Following the results in Liu et al. (2022c, Theorem 1), when the EQNTK value $Q_S^{(t)}$ is a constant, the training error decays with

$$\varepsilon_t \approx (1 - \eta Q_S^{(t)})^t \varepsilon_0 \approx e^{-\eta Q_S^{(t)} t} \varepsilon_0, \tag{15}$$

which guarantees an exponential convergence towards the global optima. To this end, the proof of Theorem 2 amounts to proving that when the number of parameters satisfies $|\boldsymbol{\theta}| = LK \gg 1$, the EQNTK can be regarded as a constant. This can be achieved by deriving an analytical solution of $Q_S^{(t)}$ on average as well as the fluctuations around the average for all iterations.

Following the above explanations, we next analyze the average of $Q_S^{(t)}$. When no confusion arises, the superscript $(t)$ of $Q_S^{(t)}$ and $U(\boldsymbol{\theta}^{(t)})$ are dropped in the subsequent analysis. By leveraging Lemmas 2 and 3, the Haar average of EQNTK yields

$$\bar{Q}_S = -\sum_{\ell=1}^{L}\sum_{k=1}^{K}\int dU_{+,\ell k}dU_{-,\ell k}\left\langle \psi_0^* \Big| (U_{+,\ell k}^*)^\dagger \left[ G_k^*, (U_{-,\ell k}^*)^\dagger H^* U_{-,\ell k}^* \right] U_{+,\ell k}^* \Big| \psi_0^* \right\rangle^2,$$

$$= -\sum_{\ell=1}^{L}\sum_{k=1}^{K}\int dU_{+,\ell k}^* dU_{-,\ell k}^* \, \text{Tr}\left( \rho_0^*(U_{+,\ell k}^*)^\dagger M_{-,\ell k} U_{+,\ell k}^* \rho_0^*(U_{+,\ell k}^*)^\dagger M_{-,\ell k} U_{+,\ell k}^* \right),$$

$$= -\sum_{\ell=1}^{L}\sum_{k=1}^{K}\int dU_{+,\ell k}^* \left( \frac{\text{Tr}^2\left( M_{-,\ell k}\right)\text{Tr}\left( (\rho_0^*)^2\right)}{d_{\text{eff}}^2 - 1} + \frac{\text{Tr}\left( (M_{-,\ell k})^2\right)\text{Tr}\left( (\rho_0^*)^2\right)}{d_{\text{eff}}^2 - 1} \right.$$

$$\left. + \frac{\text{Tr}^2\left( M_{-,\ell k}\right)\text{Tr}^2\left( \rho_0^*\right)}{d_{\text{eff}} - d_{\text{eff}}^3} + \frac{\text{Tr}\left( (M_{-,\ell k})^2\right)\text{Tr}\left( (\rho_0^*)^2\right)}{d_{\text{eff}} - d_{\text{eff}}^3} \right)$$

$$= -\sum_{\ell=1}^{L}\sum_{k=1}^{K}\int dU_{+,\ell k}^* \frac{\text{Tr}\left( (M_{-,\ell k})^2\right)}{d_{\text{eff}}^2 + d_{\text{eff}}}$$

$$= -\sum_{\ell=1}^{L}\sum_{k=1}^{K}\int dU_{+,\ell k}^* \frac{2\,\text{Tr}\left( \left( G_k^*(U_{-,\ell k}^*)^\dagger H^* U_{-,\ell k}^* \right)^2 \right) - 2\,\text{Tr}\left( (G_k^*)^2((U_{-,\ell k}^*)^\dagger H^* U_{-,\ell k}^*)^2\right)}{d_{\text{eff}}^2 + d_{\text{eff}}}$$

$$= \frac{2}{d_{\text{eff}}^2 + d_{\text{eff}}}\left( \frac{d_{\text{eff}}\,\text{Tr}((H^*)^2 - \text{Tr}^2(H^*))}{d_{\text{eff}}^2 - 1} \right)\text{Tr}\left( L\sum_{k=1}^{K}(G_k^*)^2 \right)$$

$$\approx \frac{LK\,\text{Tr}\left( (H^*)^2\right)}{d_{\text{eff}}^2}. \tag{16}$$

where the second equality employs the assumption such that $U_{-,\ell k}^*$ and $U_{+,\ell k}^*$ match the Haar distribution on the group $SU(d_{\text{eff}})$ up to the second moment, $\rho_0^* = |\psi_0^*\rangle\langle\psi_0^*|$ is the projection

of $\rho_0 = |\psi_0\rangle \langle\psi_0|$ on the subspace $V^*$, and $M_{-,\ell k} = [G_k^*, (U_{-,\ell k}^*)^\dagger H^* U_{-,\ell k}^*]$, the third equality exploits the **RTNI** package (Fukuda et al., 2019) to calculate the integration with respect to the Haar measure, the fourth equality uses the fact $\mathrm{Tr}((\rho_0^*)^2) = \mathrm{Tr}(\rho_0^*) = 1$ with $\rho_0^*$ being the pure state, the fifth equality utilizes $\mathrm{Tr}([A,B]^2) = 2\,\mathrm{Tr}(ABAB) - 2\,\mathrm{Tr}(A^2 B^2)$, the last second equality uses the **RTNI** package again, and the last equality utilizes the fact

$$\mathrm{Tr}((G_k^*)^2) = \frac{\mathrm{Tr}(G_k \Pi^* G_k \Pi^*)}{\sum_{j=1}^p \mathrm{Tr}(G_k \Pi_j G_k \Pi_j)} \, \mathrm{Tr}(G_k^2) \approx \frac{d_{\mathrm{eff}}}{2^n} \cdot 2^n = d_{\mathrm{eff}} \tag{17}$$

with $\mathrm{Tr}(G_k^2) = \mathrm{Tr}(\mathbb{I}) = 2^n$.

The fluctuation of EQNTK can be expressed as $\Delta Q_S^2 = E(Q_S^2) - \bar{Q}_S^2$, i.e.,

$$\begin{aligned}
\Delta Q_S^2 =& 2 \sum_{\ell_1, k_1 < \ell_2, k_2} \int dU_{+,\ell_1 k_1}^* dU_{+,\ell_2 k_2}^* dU_{-,\ell_1 k_1}^* dU_{-,\ell_2 k_2}^* \times \\
& \left( \begin{array}{l} \mathrm{Tr}\left( \rho_0^* (U_{+,\ell_1 k_1}^*)^\dagger M_{-,\ell_1 k_1} (U_{+,\ell_1 k_1}^*)^\dagger \rho_0^* U_{+,\ell_1 k_1}^* M_{-,\ell_1 k_1} U_{+,\ell_1 k_1}^* \right) \times \\ \mathrm{Tr}\left( \rho_0^* (U_{+,\ell_2 k_2}^*)^\dagger M_{-,\ell_2 k_2} (U_{+,\ell_2 k_2}^*)^\dagger \rho_0^* U_{+,\ell_2 k_2}^* M_{-,\ell_2 k_2} U_{+,\ell_2 k_2}^* \right) \end{array} \right) \\
& + \sum_{\ell,k} \int dU_{+,\ell k}^* dU_{-,\ell k}^* \left( \begin{array}{l} \mathrm{Tr}\left( \rho_0^* (U_{+,\ell k}^*)^\dagger M_{-,\ell k} (U_{+,\ell k}^*)^\dagger \rho_0^* U_{+,\ell k}^* M_{-,\ell k} U_{+,\ell k}^* \right) \times \\ \mathrm{Tr}\left( \rho_0^* (U_{+,\ell k}^*)^\dagger M_{-,\ell k} (U_{+,\ell k}^*)^\dagger \rho_0^* U_{+,\ell k}^* M_{-,\ell k} U_{+,\ell k}^* \right) \end{array} \right) \\
& - \bar{Q}_S^2 \\
=& \frac{LK}{d_{\mathrm{eff}}^4} \left( 8\,\mathrm{Tr}^2((H^*)^2) + 12\,\mathrm{Tr}((H^*)^4) \right) + \\
& \frac{LK}{d_{\mathrm{eff}}^5} \left( 16\,\mathrm{Tr}((H^*)^2)\,\mathrm{Tr}^2(H^*) + 48\,\mathrm{Tr}((H^*)^3)\,\mathrm{Tr}(H^*) + 40\,\mathrm{Tr}^2((H^*)^2) + \cdots \right. \\
\approx& \frac{LK}{d_{\mathrm{eff}}^4} \left( 8\,\mathrm{Tr}^2((H^*)^2) + 12\,\mathrm{Tr}((H^*)^4) \right),
\end{aligned} \tag{18}$$

where $\ell_1, k_1 < \ell_2, k_2$ refers to $\ell_1 K + k_1 < \ell_2 K + k_2$, the derivation of the second equality mainly follows the results of Liu et al. (2022b, Appendix C), and the approximation comes from the truncation and the preservation of the leading order term.

Taken together, when the number of parameters $LK \gg 1$ such that $Q_S/\Delta Q_S \approx \frac{1}{\sqrt{LK}} \ll 1$, the EQNTK can be viewed as a constant and Eqn. (15) is satisfied. $\qquad\square$

## C.1 Proof of Lemma 2

*Proof of Lemma 2.* The two conditions in the lemma, i.e., (i) any unitary $U$ in $\mathcal{U}_{\mathcal{A}}$ commutes with the Hermitian matrix $\Sigma$ and (ii) $\Sigma$ leads to the decomposition $V = \oplus_{j=1}^p V_j$ with projection $\Pi_j$, imply that $U$ is block-diagonal under the projection $\{\Pi_j\}_{j=1}^p$. In other words, we have $\Pi_{j'} U \Pi_j = 0$ for $j \neq j'$ and $\forall U \in \mathcal{U}_{\mathcal{A}}$. This observation gives the following results, i.e.,

$$\begin{aligned}
& \Pi^* U W U^\dagger \Pi^* \\
=& \Pi^* U \sum_{j=1}^p \Pi_j W \sum_{j'=1}^p \Pi_{j'} U^\dagger \Pi^* \\
=& \sum_{j,j' \in [p]} (\Pi^* U \Pi_j) W (\Pi_{j'} U^\dagger \Pi^*) \\
=& \Pi^* U \Pi^* W \Pi^* U^\dagger \Pi^* \\
=& \Pi^* U \Pi^* \Pi^* W \Pi^* \Pi^* U^\dagger \Pi^*,
\end{aligned} \tag{19}$$

where the first equality employs the fact of $\mathbb{I} = \sum_{j=1}^p \Pi_j$ and the last equality uses the property of projections $\Pi_j^2 = \Pi_j$. $\qquad\square$

## C.2 PROOF OF LEMMA 3

*Proof of Lemma 3.* The explicit form of the training error $\varepsilon(\boldsymbol{\theta}) = \langle\psi_0|\, U(\boldsymbol{\theta})^\dagger H U(\boldsymbol{\theta})\, |\psi_0\rangle - E_0$ leads to the explicit form of QNTK, i.e.,

$$
\begin{aligned}
Q &= (\nabla\varepsilon(\boldsymbol{\theta}))^\top \nabla\varepsilon(\boldsymbol{\theta}) \\
&= -\sum_{\ell=1}^{L}\sum_{k=1}^{K} \left\langle \psi_0 \Big| U_{+,\ell k}^\dagger \left[ G_k, U_{-,\ell k}^\dagger H U_{-,\ell k} \right] U_{+,\ell k} \Big| \psi_0 \right\rangle^2 .
\end{aligned} \tag{20}
$$

Similarly, for the symmetric ansatz $U(\boldsymbol{\theta})$ with the projection $\Pi^* = P P^\dagger$, the EQNTK yields

$$
\begin{aligned}
Q_S &= -\sum_{\ell=1}^{L}\sum_{k=1}^{K} \operatorname{Tr}\left( \big|\psi_0\big\rangle\big\langle\psi_0\big| U_{+,\ell k}^\dagger \left[ G_k, U_{-,\ell k}^\dagger H U_{-,\ell k} \right] U_{+,\ell k} \right)^2 \\
&= -\sum_{\ell=1}^{L}\sum_{k=1}^{K} \operatorname{Tr}\left( \Pi^* \rho_0 \Pi^* U_{+,\ell k}^\dagger \left[ G_k, U_{-,\ell k}^\dagger H U_{-,\ell k} \right] U_{+,\ell k} \right)^2 \\
&= -\sum_{\ell=1}^{L}\sum_{k=1}^{K} \operatorname{Tr}\left( \Pi^* \rho_0 \Pi^* U_{+,\ell k}^\dagger \left[ G_k, U_{-,\ell k}^\dagger O U_{-,\ell k} \right] U_{+,\ell k} \Pi^* \right)^2 \\
&= -\sum_{\ell=1}^{L}\sum_{k=1}^{K} \operatorname{Tr}\left( \Pi^* \rho_0 \Pi^* U_{+,\ell k}^\dagger \Pi^* \left[ G_k, U_{-,\ell k}^\dagger O U_{-,\ell k} \right] \Pi^* U_{+,\ell k} \Pi^* \right)^2 \\
&= -\sum_{\ell=1}^{L}\sum_{k=1}^{K} \operatorname{Tr}\left( \Pi^* \rho_0 \Pi^* U_{+,\ell k}^\dagger \Pi^* \left[ \Pi^* G_k \Pi^*, \Pi^* U_{-,\ell k}^\dagger \Pi^* H \Pi^* U_{-,\ell k} \Pi^* \right] \Pi^* U_{+,\ell k} \Pi^* \right)^2 \\
&= -\sum_{\ell=1}^{L}\sum_{k=1}^{K} \operatorname{Tr}\left( P^\dagger \rho_0 P P^\dagger U_{+,\ell k}^\dagger P \left[ P^\dagger G_k P, P^\dagger U_{-,\ell k}^\dagger P P^\dagger H P P^\dagger U_{-,\ell k} P \right] P^\dagger U_{+,\ell k} P \right)^2 \\
&= -\sum_{\ell=1}^{L}\sum_{k=1}^{K} \operatorname{Tr}\left( \rho_0^* (U_{+,\ell k}^*)^\dagger \left[ G_k^*, (U_{-,\ell k}^*)^\dagger H^* U_{-,\ell k}^* \right] U_{+,\ell k}^* \right)^2 \\
&= -\sum_{\ell=1}^{L}\sum_{k=1}^{K} \left\langle \psi_0^* \Big| (U_{+,\ell k}^*)^\dagger \left[ G_k^*, (U_{-,\ell k}^*)^\dagger H^* U_{-,\ell k}^* \right] U_{+,\ell k}^* \Big| \psi_0^* \right\rangle^2 ,
\end{aligned} \tag{21}
$$

where the second equality utilizes the assumption such that $|\psi_0\rangle$ lies in $V^*$, the third, fourth, and fifth equalities employ the property of projection operator $(\Pi^*)^2 = \Pi^*$ and Lemma 2, the final equality follows from the definitions with $|\psi_0^*\rangle = P^\dagger |\psi_0\rangle$ and $A^* = P^\dagger A P$ with $A \in \{U_{-,\ell k}, U_{+,\ell k}, G_k, H, \rho_0\}$ . $\qquad\square$

## D PROOF OF PROPOSITION 1

Before moving to elaborate on the proof of Proposition 1, we first briefly review the definition of dynamical Lie algebra (DLA).

**Definition 3** (Definition 3, Larocca et al. (2021a)). *Given an ansatz design $\mathcal{A}$, the dynamical Lie algebra (DLA) $\mathfrak{g}$ is generated by the repeated nested commutators of the operators in $\mathcal{A}$. That is*

$$
\mathfrak{g} = \operatorname{span} \langle i G_1, \cdots, i G_K \rangle_{Lie} \tag{22}
$$

*where* $\operatorname{span} \langle S \rangle_{Lie}$ *denotes the Lie closure, i.e., the set obtained by repeatedly taking the commutator of the elements in $S$.*

The proof of Lemma 1 employs the following Lemma.

**Lemma 4.** *Consider the QNN instance $(|\psi_0\rangle, U(\boldsymbol{\theta}), H)$ whose DLA is $\mathfrak{g}$. If there exists an invariant subspace $V_{\mathfrak{g}}$ including the input state $|\psi_0\rangle$ and the ground state $|\psi^*\rangle$ with dimension $d_{\mathfrak{g}}$ under $\mathfrak{g}$, then the effective dimension of this ansatz design $\mathcal{A}$ yields $d_{\mathrm{eff}} = d_{\mathfrak{g}}$.*

*Proof of Lemma 4.* We first demonstrate the equivalence between the group $\mathcal{U}_{\mathcal{A}} = \cup_{L=0}^{\infty}\{U(\boldsymbol{\theta}) : \boldsymbol{\theta} \in \mathbb{R}^{LK}\}$ and the group generated by the elements in $\mathfrak{g}$, i.e.,

$$\mathcal{U}_{\mathcal{A}} = \{e^V, V \in \mathfrak{g}\}. \tag{23}$$

$\underline{\mathcal{U}_{\mathcal{A}} \Rightarrow \{e^V, V \in \mathfrak{g}\}}$. To facilitate understanding, we consider a single-layer unitary $U(\boldsymbol{\theta})$ with $L = 1$ and the ansatz design $\mathcal{A} = \{G_1, G_2\}$. From the Baker-Campbell-Hausdorff formula, we have

$$U(\boldsymbol{\theta}) = e^{i\theta_1 G_1}e^{i\theta_2 G_2} = e^{J_1(\boldsymbol{\theta})}, \tag{24}$$

where

$$J_1(\boldsymbol{\theta}) = i\left(\boldsymbol{\theta}_1 G_1 + \boldsymbol{\theta}_2 G_2 + \frac{i\boldsymbol{\theta}_1\boldsymbol{\theta}_2}{2}[G_1, G_2] - \frac{\boldsymbol{\theta}_1^2\boldsymbol{\theta}_2}{12}[G_1, [G_1, G_2]] + \cdots\right). \tag{25}$$

Eqn. (25) implies that by merging $e^{i\boldsymbol{\theta}_1 G_1}$ and $e^{i\boldsymbol{\theta}_2 G_2}$ into a single term, the new evolution is generated by an operator $J_1(\boldsymbol{\theta})$ depending on both $\boldsymbol{\theta}_1$ and $\boldsymbol{\theta}_2$, which contains a nested commutator between $G_1$ and $G_2$. Therefore, we have $J_1(\boldsymbol{\theta}) \in \mathfrak{g}$ and $U(\boldsymbol{\theta}) \in \{e^V, V \in \mathfrak{g}\}$.

For the case of multiple layers, i.e., $U(\boldsymbol{\theta}) = \prod_{\ell=1}^{L} e^{i\boldsymbol{\theta}_{\ell_1} G_1}e^{i\boldsymbol{\theta}_{\ell_2} G_2}$, we have $U(\boldsymbol{\theta}) = e^{J_L(\boldsymbol{\theta})} \in \{e^V, V \in \mathfrak{g}\}$ by recursively applying the Baker-Campbell-Hausdorff formula to reformulate $U(\boldsymbol{\theta})$ by the $J_L(\boldsymbol{\theta}) \in \mathfrak{g}$.

$\underline{\mathcal{U}_{\mathcal{A}} \Leftarrow \{e^V, V \in \mathfrak{g}\}}$. Since each element in $\mathfrak{g}$ is a linear combination of the nested commutators in Eqn. (25), there always exists $\boldsymbol{\theta} \in \mathbb{R}^{2L}$ for any $V \in \mathfrak{g}$ such that $J_L(\boldsymbol{\theta}) = V$ and thus $e^{J_L(\boldsymbol{\theta})} = U(\boldsymbol{\theta}) \in \cup_{L=0}^{\infty}\{U(\boldsymbol{\theta}) : \boldsymbol{\theta} \in \mathbb{R}^{LK}\}$.

Taken together, we obtain Eqn. (23) in the case of $K = 2$. The results for the ansatz design $\mathcal{A}$ with more than two elements can be derived in the same manner. More details can be found in Section IV of Larocca et al. (2021b).

The equivalence of $\mathcal{U}_{\mathcal{A}}$ and $\{e^G : G \in \mathfrak{g}\}$ indicates that for any $G \in \mathfrak{g}$ and $U \in \mathcal{U}_{\mathcal{A}}$, $G$ and $U$ commutes with the same Hermitian matrix $\Sigma$ since $U$ can be expressed as $e^G$ and hence has the same Eigen-space with $G$. This implies that the invariant subspace induced by $\mathcal{U}_{\mathcal{A}}$ is the same with the one induced by $\mathfrak{g}$ and thus $d = d_{\mathfrak{g}}$. $\qquad\square$

*Proof of Proposition 1.* Following Lemma 4, the DLA-based EQNTK is the same as the EQNTK discussed in Theorem 2 because the corresponding $U(\boldsymbol{\theta})$ induces the same invariant subspace. Hence, the results achieved in Theorem 2 can be applied to the DLA-based EQNTK by replacing the effective dimension $d_{\text{eff}}$ with $d_{\mathfrak{g}}$. $\qquad\square$

## E  IMPLEMENTATION DETAILS OF THE SYMMETRIC PRUNING ALGORITHM

In this section, we elucidate Steps 2-1, 2-2, and 2-3 of the proposed SP in Alg. 1. Recall the considered problem Hamiltonian is expressed as $\widetilde{H} = H \otimes \mathbb{I}^{\otimes m}$ with $H = \sum_{j=1}^{q} \alpha_j H_j$, where $\alpha_j$ is the real coefficient and $H_j$ is the tensor product of Pauli matrices on $n$ qubits. A symmetry $S$ of a Hamiltonian $\widetilde{H}$ is a unitary operator leaving $\widetilde{H}$ invariant, i.e.,

$$S\widetilde{H}S^{\dagger} = \widetilde{H}. \tag{26}$$

All of these symmetries form a symmetry group $\mathcal{S}$ where for any two symmetries $S_1, S_2 \in \mathcal{S}$, their compositions $S_1 \circ S_2$ or $S_2 \circ S_1$ and their inverses $S_1^{-1}$ and $S_2^{-1}$ are also symmetries in $\mathcal{S}$. In SP, these symmetries are classified into three categories, namely, the system symmetry (Step 2-1), the structure symmetry (Step 2-2), and the spatial symmetry (Step 2-3). Suppose that the initialized asymmetric ansatz is $U(\boldsymbol{\theta})$, SP adopts the following methods to tailor this ansatz to obey the above symmetries.

**System symmetry.** System symmetry considers the symmetry on qubit wires. Specifically, since the problem Hamiltonian $\widetilde{H}$ can be decomposed into a tensor product of Pauli terms, the symmetry condition in Eqn. (26) holds for any unitary of the form $S_{sys} = \mathbb{I}^{\otimes n} \otimes U$, where

$U$ is an arbitrary unitary in $SU(2^m)$. All such unitaries are called the *system symmetry* and form a *subgroup* of the symmetry group $\mathcal{S}$, i.e.,

$$\mathcal{S}_{sys} = \{S_{sys} = \mathbb{I}^{\otimes n} \otimes U : U \in SU(2^m)\}.$$

The system symmetry of a unitary $V$ can be recognized if $S_{sys}V S_{sys}^\dagger = V$. With this regard, SP assigns the system symmetry to $U(\boldsymbol{\theta})$ by removing the redundant parameterized gates and the two-qubit gates associated with the last $m$ qubit wires. In doing so, the pruned ansatz has the form

$$U_{\mathrm{Pr}}(\boldsymbol{\theta}) = U_1(\boldsymbol{\theta}) \otimes \mathbb{I}^{\otimes m},$$

which yields $S_{sys}(U_1(\boldsymbol{\theta}) \otimes \mathbb{I}^{\otimes m})S_{sys}^\dagger = U_1(\boldsymbol{\theta}) \otimes \mathbb{I}^{\otimes m}$, where $U_1(\boldsymbol{\theta})$ is the unitary extracted from $U(\boldsymbol{\theta})$ (the gates applied on the first $n$ qubit wires).

**Structure symmetry.** The structure symmetry $S_{str}$ refers to the symmetry for the effective Hamiltonian $H$, which satisfies

$$S_{str}H S_{str}^\dagger = H.$$

Moreover, an ansatz $V(\boldsymbol{\theta})$ is said to be structure symmetric to the problem Hamiltonian $H$ if there exists a non-trivial symmetry $S_{str}$ (i.e., not the identity operation) and $\boldsymbol{\theta} \in \Theta\backslash\{\mathbf{0}\}$ such that $S_{str}V(\boldsymbol{\theta})S_{str}^\dagger = V(\boldsymbol{\theta})$. A feasible solution of constructing the structure symmetric ansatz is restricting the corresponding ansatz design that only contains the Pauli terms of $H$. Given the pruned ansatz $U_{\mathrm{Pr}}$ returned by Step 2-1, SP (Step 2-2) assigns the structure symmetry on it by removing specific the single-qubit gates and the two-qubit gates so that the pruned ansatz design follows $\mathcal{A} = \{H_1, \cdots, H_q\}$. The tailored ansatz returned by Step 2-2 coincides with HVA, i.e., $U_1(\boldsymbol{\theta})$ transforms to the new ansatz whose $\ell$-th layer is expressed as $\prod_{j=1}^q e^{-i\boldsymbol{\theta}_{\ell j}H_j}$ for $\forall l \in [L]$.

**Spatial symmetry.** Spatial symmetry is a discrete symmetry considering the permutation invariance for the sites of the problem Hamiltonian, which tightly relates to the problem of graph automorphism that has been widely studied in graph theory. For this reason, here we introduce the spatial symmetry from the graphical perspective and elucidate the implementations of Step 2-3 in Alg. 1. The key in this step is leveraging the algorithms developed in graph theory to automatically identify the spatial symmetry of problem Hamiltonians.

From the graphical view, an $n$-qubit Hamiltonian $H$ refers to a graph $G = (V, E)$ with $n$ vertices, where the $j$-th node $v_j \in V$ represents the $j$-th site (qubit) of $H$ and the edge $E_{i,j} \in E$ characterizes the interaction strength of the $i$-th and the $j$-th sites (qubits). This graph can further be described by an adjacency matrix $D$.

Recall that a spatial symmetry $\pi$ of a Hamiltonian $H$ is a permutation over the sites leaving $H$ invariant, i.e., $\pi H \pi^{-1} = H$ (or equivalently $[\pi, H] = 0$). In other words, the spatial symmetry $\pi$ preserves the topology invariance of $G$ such that for any $(u, v) \in E$, we have

$$(\pi(v), \pi(u)) \in E, \text{ and } \pi D \pi^{-1} = D.$$

In GSP, the action of $\pi$ on an $n$-qubit state $|\psi\rangle \to \pi|\psi\rangle$ means permuting the indices of qubits. For instance, a permutation $\pi$ with $\pi(1) = 3, \pi(2) = 1, \pi(3) = 2$ acting on the state $|\psi_1\rangle|\psi_2\rangle|\psi_3\rangle$ yields $\pi(|\psi_1\rangle|\psi_2\rangle|\psi_3\rangle) = |\psi_3\rangle|\psi_1\rangle|\psi_2\rangle$. All these permutations form a discrete group of symmetries $S_n$ with the cardinality $O(n!)$. Particularly, the spatial symmetries of the Hamiltonian is the automorphism group of its corresponding graph, defined as

$$Aut(H) = \{\pi_a \in S_n | \pi_a H \pi_a^{-1} = H\},$$

or equivalently $Aut(H) = \{\pi_a \in S_n | \pi_a D \pi_a^{-1} = D\}$. The qubits (or qubit-pairs) in the ansatz corresponding to the nodes (or edges) that can be swapped are called *equivalent qubits* (or *qubit-pairs*). More precisely, for any node (or edges) $u \in V$ (or $(u, v) \in E$), if there exists $\pi \in Aut(H)$ such that $\pi(u) = x$ (or $\pi(u, v) = (x, y)$), then the qubits (or qubit-pairs) corresponding to the node (or edge) $u$ (or $(u, v)$) and $x$ (or $(x, y)$) are called equivalent qubits (or qubit-pairs). Given the ansatz returned by Step 2-2, Step 2-3 assigns the spatial symmetry on it by correlating the single-qubit parameterized gates on the equivalent qubits or the two-qubit parameterized gates on the equivalent qubit-pairs.

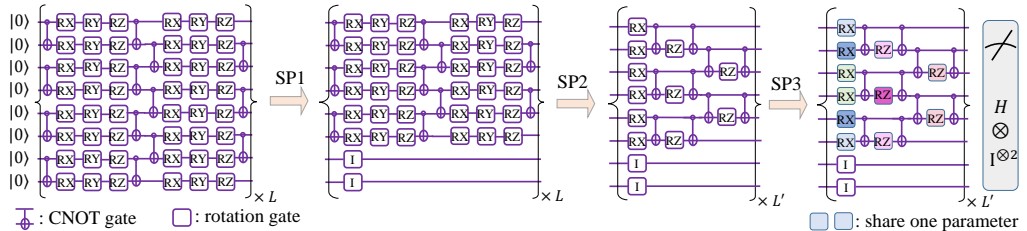

Figure 6: **Evolution of ansatz structure during symmetric pruning** From left to right shows the initial hardware efficient ansatz and ansatz structure at different stages of symmetric pruning, where 'SP1', 'SP2', 'SP3' refer to the sub-steps 2-1, 2-2, and 2-3 in Alg. 1 respectively and $L' < L$. The symbol 'RX' ('RY', 'RZ') refers to the single qubit rotation around the $x$ $(y, z)$-axis and $I$ refers to the identity gate. The rotation gates with the same color of the pruned ansatz are correlated by one individual parameter per layer.

**The flexibility of SP**. The automorphism group for the graphs corresponding to the Hamiltonians with the complicated topological structure is hard to compute manually. In this work, we employ *nauty* to automatically recognize the automorphism group of graph corresponding to the Hamiltonian *nauty* (McKay et al., 1981). Besides *nauty* , there are many heuristic algorithms to compute the automorphism group, including *Traces* (McKay & Piperno, 2014), *saucy* (Darga et al., 2004), *Bliss* (Junttila & Kaski, 2007) and *canauto* (López-Presa et al., 2014). All of them can be easily integrated into SP. Moreover, these heuristic algorithms are capable of solving most graphs for up to tens of thousands of nodes in less than a second (McKay & Piperno, 2014).

## F    THE LIMITATIONS OF APPLYING CLASSICAL PRUNING METHODS TO QNNS

Although both SP in Alg. 1 and the classical pruning techniques distill a smaller network (or an ansatz) from an over-parameterized one in the view of algorithmic implementation, the latter cannot be directly employed to enhance the power of QNNs.

Recall that a common feature of classical pruning methods is scoring each parameter or network element and then removing those accompanied with low scores. Such scores generally correspond to the magnitude of parameters (Frankle & Carbin, 2018), the gradient of parameters (Lee et al., 2018; Wang et al., 2020), and the Hessian matrix (LeCun et al., 1989; Hassibi & Stork, 1992) at the initialization stage or the phase of terminal. Unfortunately, Cerezo & Coles (2021) proved that the gradient information in QNNs with random deep ansatz exponentially vanishes with the increased number of qubits. In other words, the gradient information fails to provide any useful information to guide pruning. Meanwhile, the output of QNNs can be regarded as a periodic function of parameters (Schuld et al., 2021), which forbids employing the parameters' magnitude as the metric to guide the pruning. Therefore, it is inappropriate to straightforwardly apply classical pruning methods to QNNs, where the extracted ansatz may not promise the enhanced trainability.

In contrast with classical pruning methods, our proposal does not require any gradient information to construct the symmetric ansätze. Instead, it removes the redundant gates and shrinks the solution space according to the information of the problem Hamiltonian.

## G    MORE NUMERICAL SIMULATION DETAILS

In this section, we provide the simulation details omitted in the main text.

**Hardware efficient ansatz**. As shown in the left panel of Fig. 6, HEA yields the layer-stacking structure following Eqn. (2), where each layer consists of multiple single-qubit Pauli rotation gates and fixed two-qubit CNOT gates. In our numerical simulations, the $\ell$-th layer

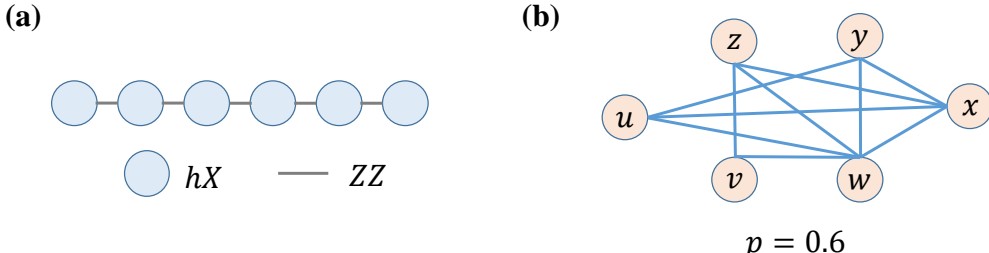

Figure 7: **Graph representation of problem Hamiltonian.** The left panel and the right panel depict the graph representation of the TFIM model and Erdos-Renyi graph with $p = 0.6$, respectively.

of the employed HEA takes the form

$$U_\ell(\boldsymbol{\theta}) = U_{ent}^{(1)} \prod_{j=1}^{n} R_X(\boldsymbol{\theta}_{j,1}^\ell) R_Y(\boldsymbol{\theta}_{j,2}^\ell) R_Z(\boldsymbol{\theta}_{j,3}^\ell) U_{ent}^{(1)}$$

$$\times U_{ent}^{(2)} \prod_{j=1}^{n} R_X(\boldsymbol{\theta}_{j,4}^\ell) R_Y(\boldsymbol{\theta}_{j,5}^\ell) R_Z(\boldsymbol{\theta}_{j,6}^\ell) U_{ent}^{(2)} \tag{27}$$

where $R_\mu(\boldsymbol{\theta}_{j,k}^\ell) = e^{-i\boldsymbol{\theta}_{j,k}^\ell \mu}$ with $\mu \in \{X, Y, Z\}$ denotes the parameterized single-qubit gate, and $U_{ent}^{(1)} = \prod_{j=1}^{\lfloor \frac{n}{2} \rfloor} \mathrm{CNOT}_{2j-1,2j}$ and $U_{ent}^{(2)} = \otimes_{j=1}^{\lfloor \frac{n-1}{2} \rfloor} \mathrm{CNOT}_{2j,2j+1}$ refer to the entangled layers with $\lfloor a \rfloor$ being the greatest integer no larger than $a$.

**Transverse-field Ising model**. A central problem in quantum many-body physics is predicting the properties of these quantum systems from the first principles of quantum mechanics. Transverse-field Ising model (TFIM) has been employed to explore many interesting quantum systems. In our numerical simulation, we employ an $n$-qubit Hamiltonian of 1D TFIM with an open boundary condition, i.e., $H_{\mathrm{TFIM}} = -\sum_{j=1}^{n-1} \sigma_j^z \sigma_{j+1}^z - \sum_{j=1}^{n} \sigma_j^x$, where $\sigma_j^\mu$ denotes the $\mu$-Pauli matrix (with $\mu = x, z$) acting on the $j$-th qubit. The effective dimension for HVA under this Hamiltonian is given by $d_{\mathrm{eff}} = n^2$ (Larocca et al., 2021a). The Hamiltonian is graphically depicted in Fig. 7(a).

**MaxCut**. Although many important problems in statistical physics and operation research (Wheeler, 2004) can be formulated as MaxCut, finding the optimal solution of MaxCut has been proven to be NP-hard (Karp, 1972) and quantum computers are expected to attain better approximated solutions than those of classical computers (Farhi et al., 2014; Zhou et al., 2022).In this work, we consider the MaxCut problem of the Erdos-Renyi graphs whose topology is less structured. An Erdos-Renyi (ER) graph on the vertex set V is a random graph in which each pair of nodes $(u, v)$ connects independently with probability $p$. Fig. 7(b) shows the instance of the ER graph used in the numerical simulation with setting $p = 0.6$.

**Evolution of ansätze**. Here we present the evolution of the ansatz structure for the transverse-field Ising model during symmetric pruning in Fig. 6, which serves as an example for better understanding the learning dynamics of our proposal. Specifically, we adopt the Hardware efficient ansatz as the initial over-parameterized ansatz, as shown in the left side of Fig. 6. The gates on the last two wires corresponding to $\mathbb{I}^{\otimes 2}$ are first removed through Step 2-1 (referred to 'SP1') in Alg. 1 to ensure the system symmetry. Subsequently, in Step 2-2, SP employs the symmetric information of problem Hamiltonian $H_{\mathrm{TFIM}}$ to remove the parameterized single-qubit gates and two-qubit gates on the first six wires such that the pruned ansatz design is $\mathcal{A}_{pr} = \{\sigma_1^x, \cdots, \sigma_6^x, \sigma_1^z \sigma_2^z, \cdots \sigma_5^z \sigma_6^z\}$. Finally, in Step 2-3, the spatial symmetry pruning correlates the parameterized gates on the equivalent qubits and qubit-pairs through the returned automorphism by the package *nauty*. In the case of TFIM, the package *nauty* returns a non-trivial automorphism $\pi(j) = n + 1 - j$ that permutes qubits from each side of the chain. This operation leads to a reduction of the number of free parameters from 11 for the ansatz pruned by 'SP2' to 6 for the ansatz returned by 'SP3'.

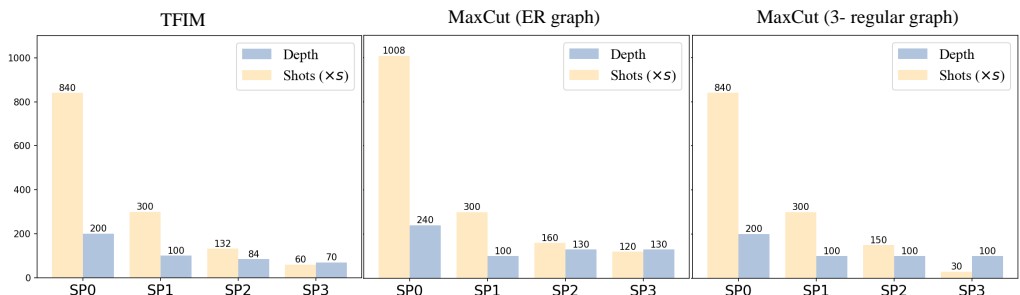

Figure 8: **The quantum resource required for achieving $\epsilon$-convergence.** The left panel, the middle panel, and the right panel depict the number of measurements required to complete one optimization step and the circuit depth required to achieve the $\epsilon$-convergence in the task of TFIM, MaxCut for Erdos-Renyi graph, and MaxCut for 3-regular graph, respectively. Particularly, the label '$\times s$' refers that the total number of shots $M$ is the product of the displayed value in the histogram and the number of shots for updating a single parameter $s$. The labels 'SP0'-'SP3' refer to the initial ansatz, the pruned ansatz after system symmetric pruning, structure symmetric pruning, and spatial symmetric pruning, respectively.

**Hardware efficiency analysis.** We use two standard metrics to quantify the hardware efficiency, i.e., (1) the number of measurements (shots) $M$ required to complete one optimization step; (2) the circuit depth $l(\epsilon)$ required to reach the over-parameterization criteria characterized by $\epsilon$-convergence with $\epsilon = 10^{-5}$. Namely, the first metric concerns the runtime cost in training QNNs, and the second metric evaluates the required quantum resources to construct the quantum circuit. To facilitate the comparison of various ansätze under the first metric, the number of shots for a single parameter update is set to be $s$, so that the total number of measurements taken to complete one optimization step linearly scales with the number of parameters, i.e., $M = LKs$. As depicted in Figure 8, in the task of TFIM, compared with the initial over-parameterized asymmetric ansatz (labeled by 'SP0'), the required number of measurements $M$ for the pruned ansatz can be dramatically reduced by 14 times; meanwhile, the required circuit depth is reduced by 3 times. In the task of Max-Cut, the hardware efficiency improvement is problem-dependent. In particular, the required circuit depth is reduced by about 1.8 times and 2 times for the Erdos-Renyi graph and the 3-regular graph, respectively. For the Erdos-Renyi graph, compared to the ansatz returned by 'SP1', the required circuit depth of the ansatz returned by 'SP2' slightly increases from 100 to 130. This increase originates from the fact that the Erdos-Renyi graph with a large number of edges requires a relatively deep circuit depth to construct the symmetric ansatz after SP1. Although the circuit depth is subtly increased, an evident benefit is a dramatic reduction of the number of measurements $M$, i.e., compared with the initial ansatz, the required $M$ for the pruned ansatz is reduced by 10 times than that of the initial ansatz. The achieved numerical results confirm that the symmetric ansatz output by our proposal can effectively improve hardware efficiency. As such, it can simultaneously reduce the required quantum resource for reaching the regime of over-parameterization, enable an efficient implementation on NISQ devices, and more importantly, improve the convergence rate so as to reduce the number of access to the quantum devices.

**Training dynamics analysis of symmetric ansatz.** Here we numerically exhibit that EQNTK has the ability to capture the training dynamics of QNNs with symmetrical ansatz. The hyperparameter settings are as follows. In the task of TFIM, the number of qubits of the Hamiltonian is set as $n = 6$. We employ the QNN with symmetric ansätze processed by SP with the number of layers $L = 80$ to optimize the loss function defined in Eqn. (1). The learning rate $\eta$ and the maximum number of iteration $T$ is set as $10^{-4}$ and 1000, respectively. Figure 9 plots the theoretically predicted residual training error according to Theorem 2, the practical residual training error $\varepsilon$ with 30 independent random initializations, and their average versus the gradient descent optimization steps. The numerical results show that the residual error $\varepsilon$ decays exponentially, which echos with the training dynamics derived in Theorem 2.

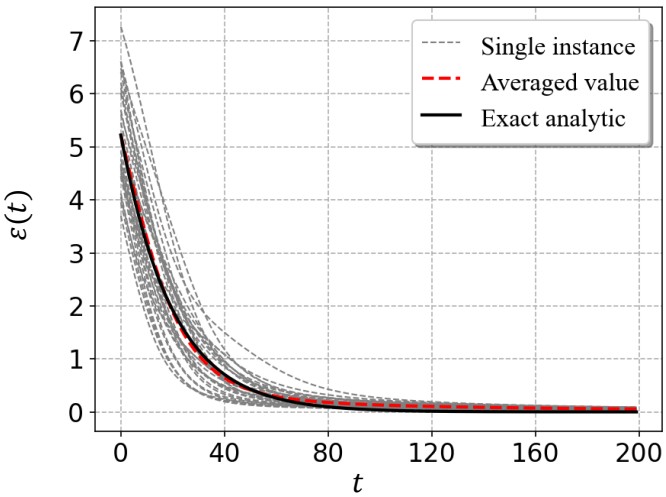

Figure 9: **Residual training error $\varepsilon$ versus the gradient descent steps $t$.** The black dotted curve, black solid curve, and red dotted curve correspond to the training dynamics of $\varepsilon(t)$ for 30 different initializations, the theoretical prediction for the average dynamics of $\varepsilon(t)$, and the numerical values for the averaged $\varepsilon(t)$, respectively.

