# OpenReview forum: "Symmetric Pruning in Quantum Neural Networks"
_ICLR.cc/2023/Conference — ICLR 2023 notable top 25%_

### Official Review · Reviewer_WF6w · 2022-10-23

**Confidence:** 4
**Correctness:** 4
**Technical Novelty And Significance:** 3
**Empirical Novelty And Significance:** 3
**Recommendation:** 8

**Clarity, Quality, Novelty And Reproducibility:**

Clarity: the paper is well organized and clearly stated in general.

Quality: this is a high-quality paper.

Novelty: the originality is high.

Reproducibility: all important parts are provided for reproducing.

**Details Of Ethics Concerns:**

None.

**Strength And Weaknesses:**

Strength:
1. The article is well-written and obtains some positive results for the training of QNNs, which is of prominent importance for researchers studying variational quantum algorithms.
2. The achieved analytical results of convergence based on the EQNTK have practical guidance on trainability without any training procedure.
3. The proposed symmetric algorithm provides a novel vision for devising a symmetric ansatz to reduce the quantum resource required for reaching the over-parameterization regime.

Weakness:
The paper did not discuss the connection with existing studies about geometric quantum machine learning or equivariant quantum circuits. I’d like to see the authors’ response to this issue.


**Summary Of The Paper:**

The article analyzes the convergence of quantum neural networks (QNNs) where the employed ansatz shares the same symmetry of the problem Hamiltonian, and gives an analytical convergence rate under the framework of the quantum neural tangent kernel. More precisely, they use the information of the problem Hamiltonian to shrink the solution space to an invariant space with dimension $d_{eff}$ (dubbed effective dimension). Furthermore, they exploit the effective dimension to propose the effective quantum neural tangent kernel (EQNTK) to capture the training dynamics symmetric ansatz. They show that a symmetric ansatz with a small effective dimension has a large EQNTK, leading to better trainability. Guided by EQNTK, they further devise a paradigm of ansatz design where a symmetric ansatz is extracted from an over-parameterized one by pruning redundant symmetry-breaking gates.

**Summary Of The Review:**

This paper provides elegant and rigorous theoretical results towards understanding the trainability of QNNs with symmetric ansatzes. The proposed algorithm is effective but still has room for further improvement.

---

> ### Author Response · Authors · 2022-11-15
> **Response**
>
> We thank Reviewer *WF6w* for the positive affirmation of our work. We notice that a minor concern of the reviewer is that the original manuscript did not discuss the connection between our works and geometric quantum machine learning or equivariant quantum circuits.
>
> - **”*The paper did not discuss the connection with existing studies about geometric quantum machine learning or equivariant quantum circuits. I’d like to see the authors’ response to this issue*.”**
>
> **Reply:** We have followed the reviewer’s suggestion to add the following words (highlighted by the red color) in the section **Related work** to discuss the connection between our works and geometric quantum machine learning or equivariant quantum circuits.
>
> *”…and utilizing results from the geometric deep learning, where the symmetry of problems comes from the training data.”*

---

### Official Review · Reviewer_1zgw · 2022-10-24

**Confidence:** 4
**Clarity, Quality, Novelty And Reproducibility:** It is a nice original paper with clea…
**Correctness:** 4
**Technical Novelty And Significance:** 3
**Empirical Novelty And Significance:** 3
**Recommendation:** 8

**Strength And Weaknesses:**

Given many recent negative results on the trainability of QNNs, such as the barren plateau and convergence, with the importance of trainability for variational quantum algorithms, results given here are useful for the researchers in this field to devise symmetric ansatz with better training performance to achieve the potential quantum advantage. The results themselves focus on the problem of ground state preparation but they are applicable to general quantum machine learning problems. Their main results of the analytical convergence rate depend on the effective quantum neural tangent kernel, which can be easily calculated by calculating the norm of parameter gradients. This facilitates the evaluation of the convergence of QNNs without any training procedure. Another contribution of this paper is proposing a novel paradigm of exploiting pruning techniques to devise symmetric ansatz. As the symmetry pattern of the problem Hamiltonian is sophisticated, they employ the automorphism algorithm from graph theory to recognize the spatial symmetry of the problem Hamiltonian and apply it to prune the ansatz.

A small collection of numerical simulations is given which show the effectiveness of symmetric pruning and justify the important role of EQNTK in guiding the trainability of QNNs. However, the paper did not numerically show that the EQNTK can capture the training dynamics of QNNs. I would like to see a figure display like Figure 2 in [1] in the author’s response.

Typos:
- In the sentence “The above results indicate that when the number of trainable parameters scales with $LK ∼ O(d_{eff}^2/(\eta Tr(H^2 )))$”, the $H$ should be $H^*$.
- “Recall the considered the problem Hamiltonian is expressed …”.

[1] Junyu Liu, Khadijeh Najafi, Kunal Sharma, Francesco Tacchino, Liang Jiang, and Antonio Mezzacapo. An analytic theory for the dynamics of wide quantum neural networks. arXiv preprint arXiv:2203.16711, 2022b


**Summary Of The Paper:**

This paper studies the trainability of quantum neural networks (QNNs) with symmetric ansatzes and how to exploit the symmetry to improve the trainability of QNNs. More precisely, it provides a theoretical understanding of the better trainability of symmetric ansatz over asymmetric ansatz by proposing the effective quantum neural tangent kernel (EQNTK) and connecting it to the over-parameterization theory. With the support of the theory, it proposes a symmetric pruning algorithm to extract a symmetric ansatz from an over-parameterized but asymmetric ansatz to improve the trainability of QNNs.

**Summary Of The Review:**

This paper obtains important theoretical results showing that constructing symmetric ansatzes improve the trainability of QNNs and provides a practical algorithm to devise symmetric ansatzes.

---

> ### Author Response · Authors · 2022-11-15
> **Response**
>
> We thank Reviewer *1zgw* for the positive affirmation of our work. We notice that a minor concern of the reviewer is that the original manuscript did not supply numerical verifications for the ability of EQNTK to capture the training dynamics of QNNs.
>
> - **”*However, the paper did not numerically show that the EQNTK can capture the training dynamics of QNNs. I would like to see a figure display like Figure 2 in [1] in the author’s response*.”**
>
> **Reply:** We have followed the reviewer’s suggestion to conduct numerical simulations as done in [1] and analyzed the corresponding training dynamics. The numerical results show that the training dynamics predicted by EQNTK can effectively capture the practical training dynamics evaluated by averaging the results of 30 different initializations. Specifically, the error in each step between the predicted and average training residual errors is below 0.15. All the numerical results are appended to Appendix G ``**More numerical simulation details’’** (highlighted by the blue color).
>
> We thank the reviewer for pointing out the typos in the manuscript, we have fixed them in the revised version.
>
> [1] *Junyu Liu, Khadijeh Najafi, Kunal Sharma, Francesco Tacchino, Liang Jiang, and Antonio Mezzacapo. An analytic theory for the dynamics of wide quantum neural networks. arXiv preprint arXiv:2203.16711, 2022b*

---

### Official Review · Reviewer_feix · 2022-11-03

**Confidence:** 2
**Correctness:** 3
**Technical Novelty And Significance:** 3
**Empirical Novelty And Significance:** 3
**Recommendation:** 8

**Clarity, Quality, Novelty And Reproducibility:**

To the best of my knowledge, this work is original. I did a quick search and it seems the authors have referenced related works adequately in the paper. The writing is good, but can be improved for non-experts in this area. In particular, I would have appreciated some primers on GSPs and Hamiltonians.

**Strength And Weaknesses:**

Strengths
+ The paper seems well-motivated and the problem statement is clearly explained.
+ The convegence bound of QNNs with various symmetric ansatz is significantly improved.

Weaknesses
- I don't think the authors did a good job at positioning their paper with prior art. For example, I would have liked a thorough comparison with related works in Section 5. Instead of having blanket statements like "Our results differ from the above literature in both theoretical and practical aspects", it would be better if the authors can clarify how this work is better in terms of both the QNN training and symmetric ansatz.
- Can the authors quantify the hardware efficiency improvement of their symmetric ansatz?

**Summary Of The Paper:**

This paper proposes a quantum NTK framework to improve the training convergence of QNNs for the GSP problem via a symmetric ansatz design with a small effective dimension. It also proposes a novel symmetric pruning algorithm to extract the symmetric ansatz from the overparameterized and assymetric ansatz. Empirical results confirm the effectiveness of this symmetric pruning algorithm and the quantum NTK framework.

**Summary Of The Review:**

This paper appears sound and novel. Please see the weaknesses listed above. I am not an expert in this area, and I am giving a rating of 6 for now. Based on the authors' rebuttal and discussions with other reviewers, I can reconsider my score.

----------
Review Update

I have changed my score to 8 based on the authors' response.

---

> ### Author Response · Authors · 2022-11-15
> **Response**
>
> We thank Reviewer *feix* for the positive assessment and helpful feedback for improving the quality of our manuscript.  We have updated our manuscript according to the reviewer’s comments (highlighted by the brown color). We hope that the following response can readily address the reviewer’s concerns.
>
> - **"*I don't think the authors did a good job at positioning their paper with prior art. For example, I would have liked a thorough comparison with related works in Section 5. ......, it would be better if the authors can clarify how this work is better in terms of both the QNN training and symmetric ansatz*."**
>
> **Reply:** Thanks for the comments. We have followed the reviewer’s advice to append a more thorough comparison with related works.  In the updated version, we emphasize that a common deficiency of prior studies in the context of **Trainability of QNNs** is the convergence analysis when the ansatz is symmetric (i.e., their convergence analysis either requires an exponential circuit depth for reaching over-parameterization or omits the factor of the circuit depth), while our work compensates for this issue by employing EQNTK to derive a tighter convergence rate (see Section **Related work**)*.*
>
> In addition, compared with prior literature in the context of **Ansätze with symmetric properties,** we clarify that the EQNTK-enabled symmetric pruning scheme is able to unearth the implicit symmetry of problem Hamiltonians whereas previous studies fail to identify them (see Section **Related work**).
>
> - ***"Can the authors quantify the hardware efficiency improvement of their symmetric ansatz?"***
>
> **Reply:** Thanks for the comments. In the revised version, we have followed the reviewer’s suggestion to quantify the hardware efficiency improvement when the symmetric ansatz is employed. To do so, we first specify two standard metrics to quantify the hardware efficiency: 1) the **total** **number of measurements** required to complete one optimization step; 2)  the **circuit depth** required to reach the over-parameterization criteria.  Namely, the first metric concerns the runtime cost in training QNNs, and the second metric evaluates the required quantum resources to construct the quantum circuit.
>
> The relevant numerical results are discussed in Section G ``**More numerical simulation details**''. In the measure of these two metrics, the achieved numerical results confirm that the symmetric ansatz output by our proposal can **effectively improve hardware efficiency.** As a result, it can simultaneously reduce the required quantum resource for reaching the regime of over-parameterization, enable an efficient implementation on NISQ devices, and more importantly, improve the convergence rate so as to reduce the number of access to the quantum devices. Concisely,  in the task of TFIM, compared with the initial over-parameterized asymmetric ansatz (labeled by 'SP0'), the required number of measurements $M$ for the pruned ansatz can be dramatically reduced by 14 times; meanwhile, the required circuit depth is reduced by 3 times. In the task of MaxCut, the hardware efficiency improvement is problem-dependent. In particular, the required circuit depth is reduced by about 1.8 times and 2 times for the Erdos-Renyi (ER) graph and the 3-regular graph, respectively. For the ER graph, compared to the ansatz returned by 'SP1', the required circuit depth of the ansatz returned by 'SP2' slightly increases from 100 to 130. This increase originates from the fact that the ER graph with a large number of edges requires a relatively deep circuit depth to construct the symmetric ansatz after SP1. Although the circuit depth is subtly increased, an evident benefit is a dramatic reduction of the number of measurements $M$, i.e., compared with the initial ansatz, the required $M$ for the pruned ansatz is reduced by 10 times than that of the initial ansatz.
>
> - **"*The writing is good, but can be improved for non-experts in this area. In particular, I would have appreciated some primers on GSPs and Hamiltonians*."**
>
> **Reply:** We agree with the reviewer’s viewpoint that adding some primers on GSPs and Hamiltonians can facilitate the non-experts in this area. As such, to improve the readability of our manuscript, in Appendix A “**Optimization of QNNs in GSP“**, we append the basics of quantum computing such as *quantum state, quantum gates*, and *quantum measurement*, and the preliminary of *Hamiltonian* and *ground state preparation*.
>
> | **Task** | **Ansatz** | **Shots ($\times s$)** | **Depth** |
> | --- | --- | --- | --- |
> | TFIM | 'SP0' | 840 | 200|
> | TFIM | 'SP1' | 300| 100|
> | TFIM | 'SP2' | 132| 84|
> | TFIM | 'SP3' | 60| 70|
> | MaxCut (ER) | 'SP0' | 1008| 200|
> | MaxCut (ER) | 'SP1' | 300| 100|
> | MaxCut (ER) | 'SP2' | 160| 130|
> | MaxCut (ER) | 'SP3' | 120| 130|
> | MaxCut (3-regular) | 'SP0' | 840| 200|
> | MaxCut (3-regular) | 'SP1' | 300| 100|
> | MaxCut (3-regular) | 'SP2' | 150| 100|
> | MaxCut (3-regular) | 'SP3' | 30| 100|

---

> > ### Comment · Reviewer_feix · 2022-11-17
> > **Review Update**
> >
> > Thanks for your response. I have updated my score to 8.

---

### Decision · Program_Chairs · 2023-01-20

**Decision:**

Accept: notable-top-25%

**Justification For Why Not Higher Score:**

The application of this work is limited to quantum neural networks which may limit its usefulness to the wider community.

**Justification For Why Not Lower Score:**

The article is well-written and obtains important results that both analyze the trainability of quantum neural networks in terms of a quantum neural tangent kernel and also devises an algorithm that improves their trainability and reduce the quantum resources needed. This paper will likely have a significant impact on the quantum neural network field.

**Metareview: Summary, Strengths And Weaknesses:**

This work proposes a quantum neural tangent kernel framework to improve the training convergence of QNNs. It provides a theoretical understanding of the better trainability of the symmetric ansatz over the asymmetric ansatz by proposing the EQNTK and connecting it to over-parameterization theory. With this, they propose a symmetric pruning algorithm to extract a symmetric ansatz from an over-parameterized asymmetric one to improve convergence speed.

Strengths:
* The work is well-motivated and the problem clearly explained.
* The analytical results of convergence based on the EQNTK have practical use on guiding trainability.
* The proposed symmetric algorithm is novel way of devising a symmetric ansatz and reducing quantum resources while improving trainability.

Weaknesses:
* These have been addressed in the author responses but mostly consisted of better positioning with previous work.

**Note From Pc:**

if the above contains the word "oral" or "spotlight" please see: "oral" presentation means -> notable-top-5% and "spotlight" means -> notable-top-25%. As stated in our emails, we are disassociating presentation type from AC recommendations